# CoNSoLe: Convex Neural Symbolic Learning

**Haoran Li**    **Yang Weng**
Arizona State University
Tempe, AZ, 85287
{lhaoran, yang.weng}@asu.edu

**Hanghang Tong**
University of Illinois Urbana-Champaign
Champaign, IL, 61820
htong@illinois.edu

## Abstract

Learning the underlying equation from data is a fundamental problem in many disciplines. Recent advances rely on Neural Networks (NNs) but do not provide theoretical guarantees in obtaining the exact equations owing to the non-convexity of NNs. In this paper, we propose Convex Neural Symbolic Learning (CoNSoLe) to seek convexity under mild conditions. The main idea is to decompose the recovering process into two steps and convexify each step. In the first step of searching for right symbols, we convexify the deep Q-learning. The key is to maintain double convexity for both the negative Q-function and the negative reward function in each iteration, leading to provable convexity of the negative optimal Q function to learn the true symbol connections. Conditioned on the exact searching result, we construct a Locally-Convex equation Learner (LoCaL) neural network to convexify the estimation of symbol coefficients. With such a design, we quantify a large region with strict convexity in the loss surface of LoCaL for commonly used physical functions. Finally, we demonstrate the superior performance of the CoNSoLe framework over the state-of-the-art on a diverse set of datasets.

## 1 Introduction

Identifying the underlying mathematical expressions from data plays a key role in multiple domains. For example, scientific discovery naturally requires learning analytical solutions to fit data. Engineering systems also need to frequently re-estimate system equations due to events, maintenance, upgrading, and new constructions [1], etc. In general, the problem, known as Symbolic Regression (SR) [2], learns the underlying equations $y = g(x)$ constructed via certain symbols, where $x$ and $y$ represent the input/output vector of the equations. If successfully learned, the equation can enjoy many important properties like high interpretability and generalizability, which will in turn significantly benefit scientific understanding and engineering planning, monitoring, and control.

Promising as it might be, SR is NP-hard [3, 4]. Mathematically, one can cast SR as an optimization problem over both discrete variables to select symbols and continuous variables to represent the symbol coefficients. Traditional solutions employ evolutionary algorithms like Genetic Programming (GP) [5]. These methods start from an initial set of expressions and continue evolving via operations like crossover or mutation. With fitness measures, GP-based algorithms can evaluate and select the best equations. However, these methods have poor scalability and limited theoretical guarantees [3].

More recent SR studies leverage Neural Networks (NNs) with high representational power. For the NN-based SR, we mainly categorize them into two groups based on the roles of the NNs. The first group employs NNs to directly model the equations, where sparsity of the NN weights is enforced to select symbols [6–10]. Thus, the problem is transformed into training the designed NN with sparsity regularization. However, due to the non-convexity of NNs, the weight selection and updating can be easily stuck in local optima, failing to find the exact equations.

The second group employs NNs to search the symbol connections, and non-linear optimizations like BFGS [11] can be employed to estimate symbol coefficients. For the searching procedure, [3, 12–14] leverage a Recursive Neural Network (RNN) as a policy network to iteratively generate optimal

36th Conference on Neural Information Processing Systems (NeurIPS 2022).

actions that can select and connect symbols. [15] employs large-scale pre-training to directly map from data to the symbolic equations. While these methods restrict the utilization of NNs in the search phase, the non-convexity of NNs can still suffer the risk of sub-optimal decisions to formulate equations. To mitigate this issue, some efforts have been made such as risk-seeking policy gradient to find the best equations [3], restricting the searching space via domain knowledge [16] and entropy regularization [13], and re-initialization [13], etc. However, they have limited theoretical guarantees.

In this paper, we propose Convex Neural Symbolic Learning (CONSOLE) with convexity under moderate conditions. To our best knowledge, we are the first to provide provable guarantees to learn the exact equations. In general, we decompose the SR problem into two sub-problems and seek convexity, respectively. In the first problem of searching symbols, we propose a double-convexified deep Q-learning to maintain the convexity of negative Q-function and negative reward functions with continuous action variables. Specifically, we use the Input Convex Neural Network (ICNN) [17] to represent both the negative Q-function and the negative reward function in each iteration. Subsequently, we prove that such a design (1) guarantees an optimal action selection at each step and (2) ensures the negative optimal Q-function, if successfully found, to be convex. Therefore, the convex negative optimal Q-function can yield global optimal decisions of equation constructions.

In the second problem of coefficient estimation, we use the search result to build a Locally-Convex Equation Learner (LOCAL) neural network. The key insight is that if the search result is correct, the loss surface of LOCAL has local regions that contain the global optima and has strict convexity. Moreover, we quantify the local regions and show the range of the region is large under mild assumptions. Therefore, initializations based on prior knowledge can often lie in the convex region, bringing the accurate coefficient estimation. Finally, we demonstrate that our CONSOLE outperforms state-of-the-art methods on a diverse set of datasets.

## 2 Related Work

**Symbolic regression using neural networks.** In addition to the review in the Introduction, there are studies treating NNs as a data augmentation tool to create high-quality data for SR [18, 2].

**Neural architecture search.** Searching the connections of LOCAL falls into the area of Neural Architecture Search (NAS). NAS tries to find an optimal architecture of a target NN with the best performance [19]. The searching algorithms can be divided into RL-based, evolutionary algorithm-based, sequential optimization-based, and gradient descent-based methods. For RL-based methods, [20] employs an RNN model to sample the architecture and utilizes the accuracy of the sampled network as the reward. [21] uses tabular Q-learning to find the connections of a target NN. However, tabular Q-learning can hardly be applicable when the state and action spaces are large, e.g., SR problem. The evolutionary algorithm employs methods such as GP [22] and tournament selection [23]. However, these methods may lack the scalability for SR problem [3]. The sequential optimization-based method is more scalable as the model complexity increases in a sequential manner [24]. Finally, the gradient descent-based method [25] builds a large and over-parameterized network to search and train. Then, regularizations like dropout are added to find the best connections. However, for these methods, the theoretical guarantees remain opaque for SR problem.

**Global optimum in neural networks.** Many studies have been conducted to seek global optimality in NNs, and they can be categorized into finding global optima for weights or input variables. The weight optimization is directly linked to finding symbol coefficients in LOCAL. Specifically, [26] investigates a single hidden layer with unbounded neurons and non-Euclidean regularization. The authors show the training can be done via convex optimization problems. [27] considers finite neurons and develops a novel duality theory to train two-layer NNs with convexity. [28] establishes a strong result that every local optimum is a global optimum for deep non-linear networks under several assumptions. [29] eliminates these assumptions and finds that with weight decay regularization (e.g., $l_2$ norm), the loss function of NN with ReLU activations is piece-wise strongly convex in local regions. However, LOCAL doesn't fit the above conditions. The second group of input variable optimization can help search for optimal inputs. [17] proposes ICNN such that the output of ICNN is convex in input variables. The key of ICNN is to restrict some weights and activation functions to preserve convexity. ICNN facilitates to design a convexified search algorithm.

## 3 Convex Neural Symbolic Learning

### 3.1 LOCAL to Hierarchically Represent the Equations and Learn Symbol Coefficients

SR problem can be decomposed into searching the symbol connections and estimating the symbol coefficients. The link of these two sub-problems is a proper representation of the underlying equation with unknown structures and weights. Then, we need to search the structures and estimate the weights. We utilize a neural network to represent the multi-input multi-output equations due to the efficiency [6]. We prove in Theorem 3 that under mild conditions and given correct structures of the NN, there are local regions in the loss surface with strict convexity. Thus, we name our NN as Locally Convex Equation Learner (LOCAL).

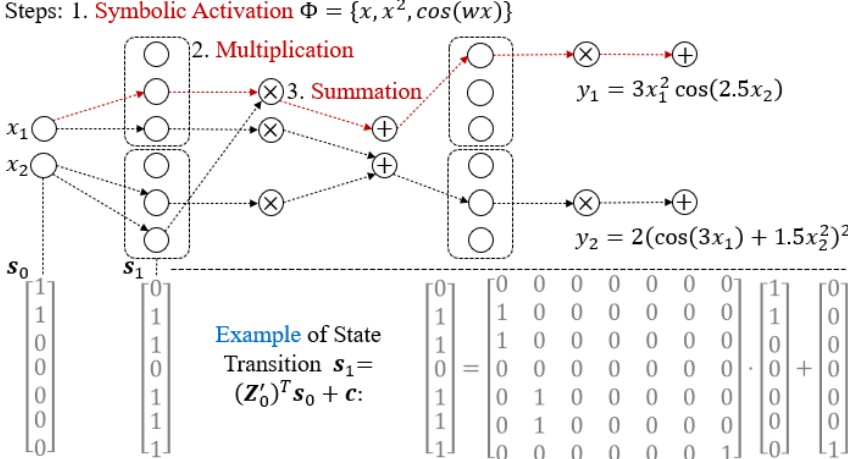

Figure 1: An example of LOCAL structure and the state transition calculation.

LOCAL should have the capacity to represent the true equation. We assume the underlying equation $y = g(x)$ follows compositionality and smoothness assumptions in [2], which are often the case in physics and many other scientific applications. Then, we build LOCAL that can be trained via input/output pairs $\{x_i, y_i\}_{i=1}^N$, where $x_i \in \mathcal{X}$ and $y_i \in \mathcal{Y}$ are the $i^{th}$ samples and $N$ is the number of data. $\mathcal{X}$ and $\mathcal{Y}$ are the input and output data spaces, respectively. By the compositionality, we propose to hierarchically map $x$ to $y$ with correct symbols. Fig. 1 shows an example of the LOCAL structure. Specifically, each input entry first goes through the activation functions from a symbol library $\Phi$. For example, in Fig. 1, we denote $\Phi = \{x, x^2, \cos(wx)\}$ to correspond with three neurons from top to bottom in the dotted black box, where $w$ is a learnable weight. As for weights of $x$ and $x^2$, they only need to appear in the later summation layer. Then, some activation outputs will be selected as multipliers for the multiplication. Subsequently, the multiplied outputs are selected and summed together. The repetition of symbolic activation, multiplication, and summation formulates final equations. For instance, the LOCAL structure in Fig. 1 can correctly represents $y_1$ and $y_2$ shown on the top right of Fig. 1.

The layer-wise connectivity and the weights of LOCAL are optimization variables for the SR problem. Mathematically, we denote $Z_k \in \{0,1\}^{n_k \times n_{k+1}}$ and $W_k \in \mathbb{R}^{n_k \times n_{k+1}}$ as the indicator and weight matrix between the $k^{th}$ and the $(k+1)^{th}$ layer of the LOCAL, respectively. $n_k$ is the number of neurons for the $k^{th}$ layers. $Z_k[i,j] = 1$ indicates that the connection exists between the $i^{th}$ neuron in the $k^{th}$ layer and the $j^{th}$ neuron in the $(k+1)^{th}$ layer, where $Z_k[i,j]$ is the $(i,j)^{th}$ entry of $Z_k$. We assume there are $(K+1)$ layers in LOCAL and denote $h_k \in \mathbb{R}^{n_k}$ to be the output of the $k^{th}$ layer. Naturally, we have $h_0 = x$ and $h_K = \hat{y}$, where $\hat{y}$ is the output of LOCAL. For the symbolic activation or summation layer, we have $h_{k+1} = Z_k^T \circ W_k^T h_k$, where $\circ$ represents the Hadamard product and helps to zero out some connections. For the multiplication layer, we have $h_{k+1}[j] = \prod_{Z_k[i,j]=1} h_k[i]$ with no weight matrix involved. In general, we denote LOCAL as $f(x; \{Z_k\}_{k=0}^{K-1}, \{W_k\}_{k=0}^{K-1})$. The search algorithm identifies $\{Z_k\}_{k=0}^{K-1}$ and the estimation process learns the corresponding weights in $\{W_k\}_{k=0}^{K-1}$ using $\{x_i, y_i\}_{i=1}^N$.

## 3.2 Search LOCAL Structures with Double Convex Deep Q-Learning

**State and action definitions in the search process.** To search the structure of LOCAL, we treat the status of each layer of LOCAL as a state and the connections between layers, i.e., $Z_k$ in LOCAL, as actions. Specifically, we denote $a_k \in \{0,1\}^{n_a}$ as the $k^{th}$ action vector where $a_k[(i-1)n_k + j] = 1$

implies that the connection $ij$ exists for the $i^{th}$ neuron in the $k^{th}$ layer and the $j^{th}$ neuron in the $(k+1)^{th}$ layer. $n_a = \max\{n_k n_{k+1}\}_{k=0}^{K-1}$ is the dimensionality of the action space. Also, we denote $\boldsymbol{s}_k \in \mathbb{Z}^{n_s}$ as the state vector to represent the current state for the $k^{th}$ layers, where $n_s = \max\{n_k\}_{k=0}^K + 1$ is the dimensionality of the state space. More specifically, the state vector $\boldsymbol{s}_k$ is composed of the values for each neuron in the $k^{th}$ layer and an entry of state index $k$. The appending of the value $k$ can avoid the duplicate state vectors for different layers. Otherwise, the duplicated states may exist and require different optimal actions, deteriorating the correct search for the true structure of LoCaL. To further quantify state vectors, we utilize a linear transformation to represent the state transition process. $\forall 0 \le k \le K - 1$, we define

$$\boldsymbol{s}_0 = [\mathbf{1}, \mathbf{0}]^T, \boldsymbol{s}_{k+1} = (\boldsymbol{Z}_k')^T \boldsymbol{s}_k + \boldsymbol{c} = \begin{bmatrix} \mathrm{Mat}(\boldsymbol{a}_k) & \mathbf{0} \\ \mathbf{0} & 1 \end{bmatrix}^T \boldsymbol{s}_k + \boldsymbol{c}, \qquad (1)$$

where the number of 1s in $\boldsymbol{s}_0$ corresponds to the input dimension. $\mathrm{Mat}(\cdot)$ is the operation to reshape a vector to a matrix, and we utilize 0s for padding to maintain the dimensions. $\boldsymbol{c} = [0, 0, \cdots, 1]^T$ is a constant vector to increase the entry of the state index from $k$ to $k + 1$. We show this linear state transition is essential to guarantee the convexity of negative optimal Q-function in Theorem 1. For the calculation example of state transition, one can refer to Fig. 1. Then, we can obtain $\boldsymbol{Z}_k$ from $\mathrm{Mat}(\boldsymbol{a}_k)$ by deleting the filled 0s. Then, the search will always start at $\boldsymbol{s}_0$ and end at $\boldsymbol{s}_K$ in one episode. Thus, we define a trajectory as a sequence of searched state-action pairs for $\{(\boldsymbol{s}_k, \boldsymbol{a}_k)\}_{k=0}^{K-1}$. This trajectory formulates $\{\boldsymbol{Z}_k\}_{k=0}^{K-1}$ in the LoCaL function $f(\boldsymbol{x}; \{\boldsymbol{Z}_k\}_{k=0}^{K-1}, \{\boldsymbol{W}_k\}_{k=0}^{K-1})$.

The above states, actions, state transitions, initial state $\boldsymbol{s}_0$, and a discount factor $\gamma$ can form a Controlled Markov Process (CMP) [30, 31], which is a Markov Decision Process (MDP) without a reward function [31]. In the following part, we define our reward function based on an end-of-trajectory reward [30]. Although the reward function is non-Markov, we prove in Theorem 1 that under our settings, there exists an optimal Q-function. The proof can be seen in Appendix A.3.

**Double convex deep Q-learning to search optimal actions.** To optimize over the defined CMP for optimal actions, we seek certain convexity with provable optimal results. Thus, we propose a double convex deep Q-learning with the convex negative reward function and negative value function (i.e., Q-function). Based on Bellman equations [32], this design will ensure the convexity of the negative optimal Q-function and global optimal solutions. More proof details can be referred to Theorem 1 and Appendix A.3.

Specifically, we utilize an Input Convex Neural Network (s) [17] to model the reward function $-R(\boldsymbol{s}_k, \boldsymbol{a}_k)$ such that $-R(\boldsymbol{s}_k, \boldsymbol{a}_k)$ is convex in states and actions. $-R(\boldsymbol{s}_k, \boldsymbol{a}_k)$ requires proper training to do the correct evaluation. In the $t^{th}$ episode, we collect the $t^{th}$ trajectory sample $\{(\boldsymbol{s}_k^t, \boldsymbol{a}_k^t)\}_{k=0}^{K-1}$. Then, the output can be defined as the end-of-trajectory reward to evaluate the obtained LoCaL which is denoted as $f_t(\boldsymbol{x}; W_t)$, where $W_t$ is the weight set of LoCaL. Basically, we utilize gradient method (e.g., Adam [33]) to train $f_t(\boldsymbol{x}; W_t)$ and obtain the optimal set of weights $W_t^*$ for the $t^{th}$ episode of LoCaL by minimizing the Mean Square Error (MSE). Then, we can calculate the Normalized Root-Mean-Square Error (NRMSE) [3] of the trained $f_t(\boldsymbol{x}; W^*)$ such that $\mathrm{NRMSE}_t = \frac{1}{\sigma_y}\sqrt{\frac{1}{N}\sum_{i=1}^N (\boldsymbol{y}_i - f_t(\boldsymbol{x}_i; W_t^*))^2}$, where $\sigma_y$ is the standard deviation of the outputs. The output of the reward function can be calculated as $R_t = \frac{1}{1+\mathrm{NRMSE}_t}$. Therefore, we can train $-R(\cdot)$ using $\{\{\boldsymbol{s}_k^t, \boldsymbol{a}_k^t\}_{k=0}^{K-1}, -R_t\}$.

Although training $R(\cdot)$ utilizes the samples of discrete states and actions, we aim to solve the continuous convex optimization for optimal sequential decisions. Thus, we first show that $R(\cdot)$ can be defined over the continuous action space. By definitions of the discrete actions, we restrict the continuous action space in a hypercube $\mathrm{conv}(\{0, 1\}^{n_a})$, i.e., a convex hull of the discrete actions. Then, the discrete actions are the endpoints of the hypercube. Notably, it is doable to utilize a continuous convex function to fit these endpoints with end-of-trajectory rewards. Especially, the strictly minimal reward value $-R_t$ only lies in the endpoint that guarantees the correct structure of LoCaL. Therefore, one can utilize piece-wise linear functions with convexity to represent $R(\cdot)$ over $\mathrm{conv}(\{0, 1\}^{n_a})$ and ensure one endpoint has the minimal value. The piece-wise linearity can be achieved via using ReLu activations to $R(\cdot)$.

With the defined continuous action space, we utilize another ICNN to represent $-Q(\boldsymbol{s}_k, \tilde{\boldsymbol{a}}_k)$ such that $-Q(\boldsymbol{s}_k, \tilde{\boldsymbol{a}}_k)$ is convex in the continuous action $\tilde{\boldsymbol{a}}_k \in \mathrm{conv}(\{0, 1\}^{n_a})$ given fixed $\boldsymbol{s}_k$. To update

Q values, we have the following iterative computations based on the temporal difference [21]:

$$Q_{t+1}(\boldsymbol{s}_k, \tilde{\boldsymbol{a}}_k) = Q_t(\boldsymbol{s}_k, \tilde{\boldsymbol{a}}_k) + \alpha\big(R(\boldsymbol{s}_k, \tilde{\boldsymbol{a}}_k) + \gamma \max_{\tilde{\boldsymbol{a}}} Q_t(\boldsymbol{s}_{k+1}, \tilde{\boldsymbol{a}}) - Q_t(\boldsymbol{s}_k, \tilde{\boldsymbol{a}}_k)\big), \qquad (2)$$

where $Q_t(\boldsymbol{s}_k, \tilde{\boldsymbol{a}}_k)$ is the Q value at the $t^{th}$ episode for state $\boldsymbol{s}_k$ and action $\tilde{\boldsymbol{a}}_k$. $\alpha$ and $\gamma$ are pre-defined learning rate and discount factor, respectively. Therefore, one can solve a convex optimization problem $\max_{\boldsymbol{a}} Q_t(\boldsymbol{s}_{k+1}, \tilde{\boldsymbol{a}})$ to obtain the (approximately) global optimal action for Equation (2) in each iteration. Thus, the optimization problem is:

$$\tilde{\boldsymbol{a}}^* = \arg\min_{\tilde{\boldsymbol{a}}} -Q(\boldsymbol{s}, \tilde{\boldsymbol{a}}), \tilde{\boldsymbol{a}} \in \text{conv}(\{0,1\}^{n_a}). \qquad (3)$$

Based on [17], this convex optimization problem can be solved via a bundle entropy method. After obtaining a continuous solution $\tilde{\boldsymbol{a}}^*$, we discretize it to a discrete vector $\boldsymbol{a}^*$ to build LOCAL. One simple method is to enforce $\boldsymbol{a}^*[i] = 1$ if $\tilde{\boldsymbol{a}}^*[i] \geq 0.5$, and otherwise $\boldsymbol{a}^*[i] = 0$. Thus, both $Q(\boldsymbol{s}_k, \tilde{\boldsymbol{a}}_k)$ and $Q(\boldsymbol{s}_k, \boldsymbol{a}_k)$ can be trained using Equation (2). Practically, we introduce $\epsilon$-greedy strategy [34], experience replay [35] and a target Q-network [36] to update Q-function, thus boosting the convergence to the optimal Q-function. The overview of our framework is in Appendix A.1, Algorithm 1. The specific algorithm is in Appendix A.2, Algorithm 2. Finally, by Theorems 1-2, the discrete optimal actions can generate the correct structures of LOCAL and exact equations.

**Symbolic static and dynamic constraints.** Constraints can be added to accelerate the search process [3, 2]. For example, in Algorithm 2, we propose a constraint checking program for the state-action pairs to avoid the invalid search, suitable for arbitrary restrictions. Then, we emphasize a general type of constraint, symbolic constraint, for the SR problem. The symbolic static constraint requires that each equation contains only a subset of symbols. For example, the equation $y_1 = x_1 x_2 \cdots x_{100}$ shouldn't exist since it is too complicated for real-world systems. This constraint eliminates part of the action space and can be checked by counting the number of 1s in the action vector. The symbolic dynamic constraint can gradually reduce the search space based on symbol correlations. Specifically, we investigate the $(K-1)^{th}$ layer's neurons that are linearly summed to form the equation. If some of these neurons have strong linear correlations to the output neuron (e.g., Pearson correlation coefficient larger than 0.99), they should be kept subsequently. Namely, we can maintain the path from input neurons to the neurons to be kept and reduce the search space.

## 4 Theoretical Analysis

We employ explorations, experience replay, and a target Q-network to boost the convergence to the optimal Q function. However, our extra requirement of the convex shape for the negative Q-function may deteriorate the convergence performance. Thus, we first prove that in CONSOLE, the negative optimal Q-function is also convex so that the convex design doesn't affect the convergence. Then, we prove that the convexity of the negative optimal Q-function eventually yields the exact equations.

**Theorem 1.** $\forall 0 \leq k \leq K-1$, the negative optimal Q-function $-Q^*(\boldsymbol{s}_k, \tilde{\boldsymbol{a}}_k)$ in the proposed CONSOLE framework exists and is convex in $\boldsymbol{s}_k$ and $\tilde{\boldsymbol{a}}_k$, where $\boldsymbol{s}_k$ is the discrete state and $\tilde{\boldsymbol{a}}_k$ is the continuous action at the $k^{th}$ stage.

The proof can be seen in Appendix A.3. Based on the convexity of negative optimal Q-function, the optimal sequence of states $(\boldsymbol{s}_0, \boldsymbol{s}_1^*, \cdots, \boldsymbol{s}_K^*)$ and actions $(\boldsymbol{a}_0^*, \boldsymbol{a}_1^*, \cdots, \boldsymbol{a}_{K-1}^*)$ can be found via solving the convex optimization problem. Then, we have the following theorem.

**Theorem 2.** Let $f^*(\cdot; W)$ denote the LOCAL constructed by the optimal sequences of states $(\boldsymbol{s}_0, \boldsymbol{s}_1^*, \cdots, \boldsymbol{s}_K^*)$ and actions $(\boldsymbol{a}_0^*, \boldsymbol{a}_1^*, \cdots, \boldsymbol{a}_{K-1}^*)$ from $-Q^*(\cdot)$, where $W$ is the set of weights of $f^*(\cdot; W)$. If $f^*(\cdot; W)$ can be trained with noiseless datasets and the training can achieve the global optimal weights $W^*$, $f^*(\cdot; W^*)$ can be simplified to the true equation $g(\cdot)$.

For the optimal search structure of LOCAL, i.e., $f^*(\cdot; W)$, the optimal weight set $W^*$ is also trained via gradient method like Adam [33]. The proof can be seen in Appendix A.4. Theorem 2 requires that LOCAL can learn the global optimal weights. The requirement is generally hard to achieve due to the non-linearity and non-convexity of LOCAL. However, we show that with mild assumptions, there are local regions in the LOCAL loss surface with strict convexity. Then, if we have proper initializations, the gradient-based weight updating can find the global optimum. Specifically, we have the following theorem.

**Theorem 3.** *Assume the following conditions hold:* (1) *the equation $g(\boldsymbol{x})$ is $C^2$ smooth and has bounded second derivatives with respect to weights,* (2) $\exists \boldsymbol{x} \in \mathcal{X}$, *$g(\boldsymbol{x})$ has non-zero gradients with respect to weights,* (3) *the structure of* LOCAL *is correctly searched to exactly represent symbols and symbol connections in $g(\boldsymbol{x})$, and* (4) *the training dataset of* LOCAL *is noiseless. Then, for the MSE loss surface of* LOCAL*, each global optimal point has a strictly convex local region.*

The proofs can be seen in Appendix A.5. Note that Assumptions 1-2 easily hold for common physical equations in nature [2]. Assumption 3 relies on the search algorithm, and we show, both theoretically and numerically, that our double convex deep Q-learning has good performances. Assumption 4 relies on the quality of data and we focus on the noiseless data in this paper. What's more, Equation (6) in the proof suggests that if the absolute noise values are small, the locally convex region still exists. We also numerically prove CONSOLE is robust under certain noise levels in Section 5.5. To summarize, these assumptions are acceptable. To quantify the range of the local region for a LOCAL with a certain complexity, we have the following theorem.

**Theorem 4.** *Suppose Assumptions 1-4 in Theorem 3 hold. For a* LOCAL *with one symbolic activation, multiplication, and summation layer, the set of local convex regions with global optima is*
$$U = \{W \Big| \frac{\left| \frac{d}{dt}\big|_{t=0} \hat{y}(\boldsymbol{x}_i, W+tX) \right|^2}{\eta \left| \frac{d}{dt}\big|_{t=0} \hat{y}(\boldsymbol{x}_j, W+tX) \right|} > |\hat{y}(\boldsymbol{x}_k, W) - y_k| \}, \text{ where notations are defined in the proof.}$$

The proofs can be seen in Appendix A.6. We explain the region range is large for a stable system that satisfies all assumptions in Theorem 4.

**Physical interpretations of the convex region size.** Physical systems in scientific and engineering domains have certain stability that can withstand parameter changes to some extent. Further, this ability should hold for arbitrary $\boldsymbol{x} \in \mathcal{X}$ and $\boldsymbol{y} \in \mathcal{Y}$. Thus, we can assume $\frac{d}{dt}\big|_{t=0}\hat{y}(\boldsymbol{x}_i, W+tX) \approx \frac{d}{dt}\big|_{t=0}\hat{y}(\boldsymbol{x}_j, W+tX) \approx \frac{d}{dt}\big|_{t=0}\hat{y}(\boldsymbol{x}_k, W+tX)$ for $\boldsymbol{x}_i, \boldsymbol{x}_j, \boldsymbol{x}_k \in \mathcal{X}$. Then, the inequality in set $U$ can be approximately rewritten as $\frac{1}{\eta} > ||\boldsymbol{w} - \boldsymbol{w}^*||_2$, where $\boldsymbol{w}$ and $\boldsymbol{w}^*$ are the vectorized $W$ and $W^*$, respectively. Namely, the distance between any point $W$ in the region to the global optimal point $W^*$ in the region is bounded by $\frac{1}{\eta}$. Based on Equation (19), $\eta$ is the bound of the ratio of second derivative to the first derivative. For a stable system, this ratio should be small. Otherwise, the system can easily crash with a small parameter disturbance. Thus, $\frac{1}{\eta}$ is relatively large and so is the region of $U$. An example of the range is displayed in Section 5.2. Finally, the above analysis also holds when the LOCAL of the system equation has more than one symbolic activation, multiplication, and summation layers. This is because Equation (20) always holds as long as we can find a $\eta$ to bound the ratio of the second derivative to the first derivative, which is irrelevant to the structure of LOCAL.

## 5 Experiments

### 5.1 Settings

**Datasets.** We use the following datasets for testing. (1) **Synthetic datasets**. We create two datasets, $\text{Syn}_1$ and $\text{Syn}_2$, for testing. $\text{Syn}_1$ has the following equations: $y_1 = 3x_1^2 \cos(2.5x_2)$, $y_2 = 4x_1x_3$, and $y_3 = 3x_3^2$. $\text{Syn}_2$ is more complex with the following equations $y_1 = \sqrt{2.2x_1}x_2 + x_1x_2^2$, $y_2 = \sin(1.8x_1)\big(\log(3x_2) + \sqrt{x_3}\big)$, $y_3 = \sqrt{3.7x_3}\log(1.6x_1) + x_1^2$. For the training data, each input variable is randomly sampled from a uniform distribution of $U(1, 2)$ to avoid invalid values like $\log(0)$. Totally, we create $2,000$ samples for training. Then, in the test phase, we utilize another $2,000$ samples whose input variables are sampled from $U(3, 4)$. The symbolic activation pools are $\{x, x^2, \cos(x)\}$ and $\{\sqrt{x}, x, x^2, \log(x), \sin(x)\}$ for $\text{Syn}_1$ and $\text{Syn}_2$, respectively. (2) **Power system dataset.** Power flow equation determines the operations of electric systems [37]. For node $i$ in an $M$-node system, the equation can be written as $p_i = \sum_{m=1}^{|M|} G_{im}(u_iu_m + v_iv_m) + B_{im}(v_iu_m - u_iv_m)$ and $q_i = \sum_{m=1}^{M} G_{im}(v_iu_m - u_iv_m) - B_{im}(v_iu_m - u_iv_m)$, where $u_i$ and $v_i$ are the real and imaginary components of the voltage phasor at node $i$. $p_i$ and $q_i$ are the active and reactive power at node $i$. $G_{im}$ and $B_{im}$ represent the physical parameters of line $im$. If line $im$ does not exist, $G_{im} = B_{im} = 0$. Therefore, we can treat $\boldsymbol{x} = [u_1, v_1, \cdots, u_M, v_M]^T$ and $\boldsymbol{y} = [p_1, q_1, \cdots, p_M, q_M]^T$. The target is to learn the underlying system topology and parameters, which has broad impacts on the power domains [38]. In this experiment, we implement simulation from a 5-node system using MATPOWER [39] and two year's hourly data. The first $8,760$ points are used for training while the remaining samples are used for testing. The symbolic activation pool is $\{x\}$. We denote this dataset as

Pow. (3) **Mass-damper system dataset.** Equations of the mass-damper system can be written as: $\dot{q} = -DRD^\top M^{-1} q$, where $\dot{q}$ is a vector of momenta, $D$ is the incidence matrix of the system, $R$ is the diagonal matrix of the damping coefficients for each line of the system, and $M$ is the diagonal matrix of each node mass of the system. Thus, we can set $y = \dot{q}$ and $x = q$ and the goal is to learn the parameter matrix $-DRD^\top M^{-1}$. We conduct the simulation via MATLAB for a 10-node system and obtain $6,000$ points for 1min simulation with a step size to be $0.01$s. The first $3,000$ samples are used for training while the rest samples are used for testing. The symbolic activation pool is $\{x\}$. Then, we denote the dataset as Mas.

**Benchmark methods**. The following benchmark methods are utilized. (1) Deep Symbolic Regression (**DSR**) [3]. DSR develops an RNN-based framework to search the expression tree. Especially, the risk-seeking policy gradient is utilized to seek the best performance. Then, BFGS [11] can solve the non-linear optimization and estimate the symbol coefficients. (2) Vanilla Policy Gradient (**VPG**) [3]. VPG is a vanilla version of DSR with a normal policy gradient rather than the risk-seeking method. (3) Equation Learner (**EQL**) [6, 7]. EQL creates an end-to-end NN to select symbols and estimate the coefficients. The sparse regularization is enforced for the NN weights to search symbols. (4) Multilayer Perceptron (**MLP**). We also employ a standard MLP to learn the regression from $x$ to $y$. We only evaluate the extrapolation capacity of MLP in the test dataset. For DSR, VPG, and EQL methods, based on the input datasets, we adjust the symbol and operator library to enable the same searching space as CONSOLE for fair comparisons. We run the benchmark methods 5 times with different random seeds and present their best results. As for our method, we only run 1 time and obtain good results due to the convex design and the $\epsilon$-greedy strategy.

**Metrics for evaluation.** We employ the following metrics. (1) Average coefficient estimation percentage error $E_c$. For an equation with $H$ symbols, We calculate the error as $E_c = \frac{1}{H}\sum_h \mathrm{PE}(w_h, \hat{w}_h)$, where $w_h$ and $\hat{w}_h$ represents the true and the estimated coefficients for the $h^{th}$ symbol, respectively. PE is the operation to calculate the percentage error. If there is no matched symbol for the $h^{th}$ true symbol, we denote $\mathrm{PE}(w_h, \hat{w}_h) = 100\%$. Note that when calculating $E_c$, proper simplifications may be needed. For example, $\cos\big(2.5(\sqrt{x})^2\big) = \cos(2.5x)$. (2) NRMSE in the test dataset. We measure the extrapolation capacity in the test dataset and utilize NRMSE employed in Section 3.2. Finally, the hyper-parameter settings can be seen in Appendix A.7.

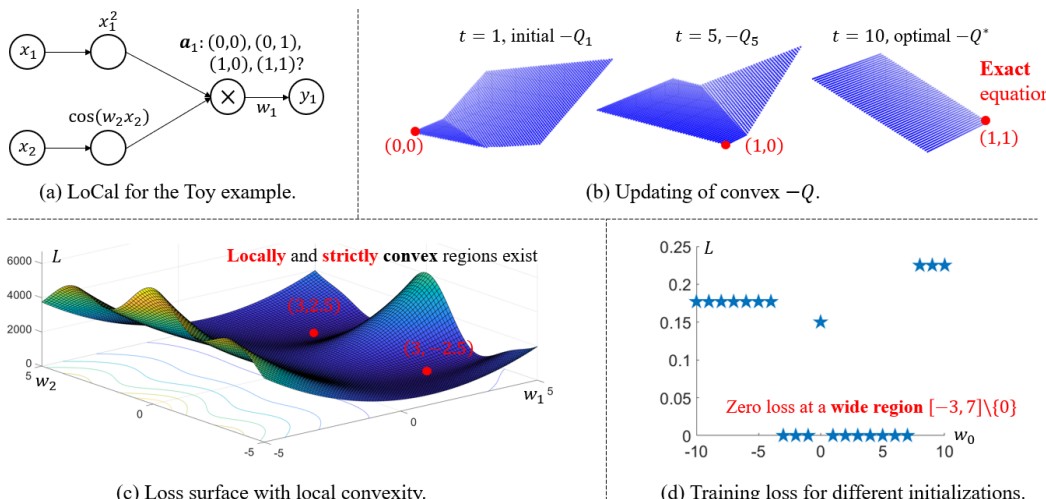

(a) LoCal for the Toy example.

(b) Updating of convex $-Q$.

(c) Loss surface with local convexity.

(d) Training loss for different initializations.

Figure 2: Illustrations of convex mechanisms using a toy example.

## 5.2 Verification and 3-D Visualization of Convex Mechanisms

We first utilize a toy example to verify the benefits of convex designs for two sub problems. Specifically, we consider to learn $y_1 = 3x_1^2 \cos(2.5x_2)$ with the loss function $L = \sum_i^N \big(y_i[1] - w_1 x_i[1]^2 \cos(w_2 x_i[2])\big)^2$, where the data sample notations are defined in Section 3.1. As shown in Fig. 2a, we design a 4-layer LOCAL with two learnable parameters $w_1$ and $w_2$ to represent the coefficients. Thus, in the search phase, the goal is to identify the action $a_1$. We plot $-Q_t(\cdot)$ when $t = 1, 5, 10$ in Fig. 2b. The convexity always exists so that the algorithm can quickly find the global optimal solutions in red dots. Next, as the agent update Q values with true rewards, the Q-function

converges to the optimal function within 10 episodes. Finally, the convex $-Q^*(\cdot)$ remains unchanged and can bring the optimal action and the true equation, which supports Theorem 2.

Subsequently, we plot the loss surface of $L$ in Fig. 2c. We find that around the two global optima $(3, 2.5)$ and $(3, -2.5)$, there are convex regions that have an approximate quadratic shape. To further quantify the range for proper initialization, we vary the initial weight $w_0 \in \{-10, -9, \cdots, 10\}$ for $w_1$ and $w_2$ in LOCAL. Fig. 2 reports the final training loss with respect to different $w_0$, and we find the safe range for initialization is $[-3, 7] \setminus \{0\}$. This range is relatively large when compared to the optimal values. These observations support our Theorems 3 and 4. Finally, $w_0 = 0$ does not work since $\frac{\partial L}{\partial w_2}\big|_{w_2=0} = 0$ always holds, which prevents the weight updating using gradient methods.

Table 1: The learned equations for $\text{Syn}_1$ and $\text{Syn}_2$.

| | CONSOLE | DSR |
|---|---|---|
| $\text{Syn}_1$ | $y_1 = 3x_1^2 \cos(2.5x_2)$ 
 $y_2 = 3.999x_1x_3$ 
 $y_3 = 3x_3^2 \cos(0.005x_1)$ | $y_1 = 0.905x_2 + 3.88\cos(2.48x_2) + 1.74x_1^2\cos(1.98x_1)$ 
 $y_2 = 4.02x_1x_3$ 
 $y_3 = 3x_3^2$ |
| $\text{Syn}_2$ | $y_1 = 1.48\sqrt{x_1}x_2 + 1.00x_1x_2^2$ 
 $y_2 = \sin(1.8x_1)\big(\log(2.999x_2) + 0.999\sqrt{x_3}\big)$ 
 $y_3 = 1.933\sqrt{x_3}\log(1.598x_1) + 1.002x_1^2$ | $y_1 = 1.223\sqrt{x_1}x_2 + 0.181x_1\log(x_3) + 0.925x_1x_2^2$ 
 $y_2 = \sin(1.633x_1)\log(2.965x_2)$ 
 $+0.874\sin(1.723x_1)\sqrt{x_3}$ 
 $y_3 = 2.081\sqrt{x_3}x_2 + 1.045x_1^2$ |
| | VPG | EQL |
| $\text{Syn}_1$ | $y_1 = -0.364x_1^2\cos(1.56x_3)$ 
 $+4.707x_2 + 0.854x_2^2\cos(1.98x_1)$ 
 $y_2 = 3.293x_2 + 0.554\cos(2.82x_2)$ 
 $y_3 = 3x_3^3$ | $y_1 = 0.23x_1 + 0.021x_3^2 + 0.283x_3$ 
 $y_2 = 0.03x_1x_3 + 0.488x_1$ 
 $+0.045x_3^2 + 0.6x_3$ 
 $y_3 = 0.366x_1 + 0.03x_3^2 + 0.45x_3$ |
| $\text{Syn}_2$ | $y_1 = 1.462\log(x_1)x_2 + 0.830x_1x_2^2$ 
 $y_2 = \sin(1.220x_1)\log(3.024x_2)$ 
 $+0.248\sin(1.454x_1)x_2^2 + 0.567\sin(1.56x_3)$ 
 $y_3 = 2.081\sqrt{x_3}x_2 + 1.045x_1^2$ | $y_1 = 0.44x_2 + 0.2x_1^2 + 0.14x_1x_2^2 + 0.45x_1x_2$ 
 $+0.51x_1 + 0.24x_2^2 + 0.55x_2 + 0.705$ 
 $y_2 = 0.018x_1^2 + 0.012x_1x_2^2 + 0.0636$ 
 $y_3 = 0.383x_2 + 0.357x_3 + 0.31x_1x_2 + 0.487$ |

## 5.3 Convexity Guarantees of CONSOLE to Learn Correct Equations

In this subsection, we report the results of the equation learning. First, we list the learned equations for $\text{Syn}_1$ and $\text{Syn}_2$ in Table 1. Table 1 presents that CONSOLE has the best performance in most of the equations while DSR ranks second. In particular, CONSOLE can accurately learn all equations in $\text{Syn}_1$ and $\text{Syn}_2$. The superior performance is mostly due to the convex design of the search and the coefficient estimation process with provable guarantees. In addition, we observe that for the result of CONSOLE in learning $y_3$ of $\text{Syn}_1$, we have $3x_3^2\cos(0.005x_1) \approx 3x_3^2$. This shows that there is a possibility that the search result of CONSOLE might not be optimal (i.e., an extra consine term exists but is close to 1), but the learned equation is still highly accurate. Such an observation nonetheless guides further study of the coupling relationship between the search and the estimation procedures.

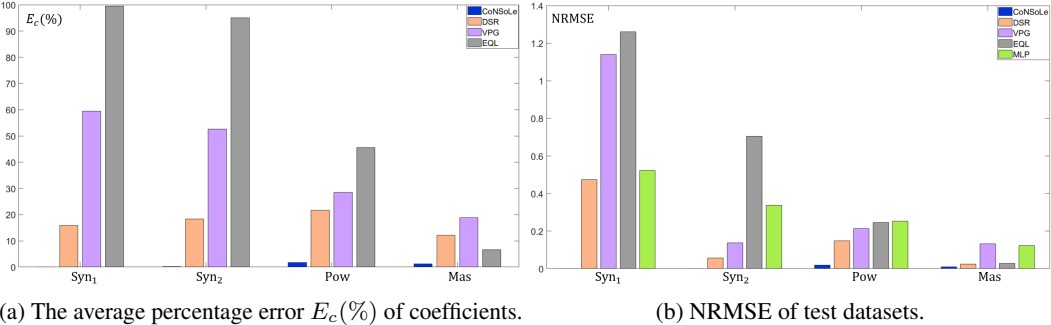

(a) The average percentage error $E_c(\%)$ of coefficients.      (b) NRMSE of test datasets.

Figure 3: Results of equation learning for different methods and datasets.

DSR doesn't perform well when the underlying equation is relatively complex, e.g., $y_1$ in $\text{Syn}_1$ and $y_1$ and $y_2$ in $\text{Syn}_2$. This is because DSR may still fall into a local optimal solution despite risk-seeking policy design. VPG method performs worse than DSR since VPG considers an expected reward [3]. Finally, EQL performs the worst as it merges the symbol search and the coefficient estimation in one NN model, which provides few guarantees of accurate learning. The above observations and analyses

are consistent with the result of coefficient estimation errors and prediction errors in Fig. 3a and 3b. For the Pow and Mas, CONSOLE doesn't learn completely exact equations within $T = 600$ episodes. This is because they have a large number of variables to be considered. However, CONSOLE is still better than other methods.

### 5.4 Ablation Study: Exploration and Convex Search are Essential

We conduct an ablation study to further understand what factors are important in the CONSOLE. We test the result with $\mathrm{Syn}_1$ and $\mathrm{Syn}_1$ and report the $E_c(\%)$ values. Specifically, we investigate the following cases. (1) No ablation. (2) Drop exploration in deep Q-learning. We delete the $\epsilon$-greedy strategy. (3) Drop double-convex deep-Q learning. We replace this design with a traditional deep-Q learning. (4) Drop coefficient estimation using LOCAL. After learning the structure of LOCAL, we reformulate a non-linear optimization and utilize BFGS [11] in DSR, instead of gradient descent in LOCAL, to estimate the coefficient. (5) Drop static symbolic constraint. (6) Drop dynamic symbolic constraint. These two constraints are mentioned in Section 3.2. Then, Fig. 4 shows that cases (2) and (3) cause large errors. For case (2), if no exploration strategy is added, the updating of the Q-function and the reward function is slow. For case (3), the non-convex search induces many sub-optimal actions in the search process. Thus, these two cases cause a slow search process and significant errors after $T = 600$ episodes. For symbolic constraints in (5) and (6), removing them increases the error for $\mathrm{Syn}_2$ and $\mathrm{Syn}_1$, respectively. This shows these constraints are beneficial to the search process. Finally, we find that utilizing BFGS in (4) can bring good results with initialization in the locally convex region. Since the non-linear optimization has the same loss surface as LOCAL, the locally convex region in Theorems 3 and 4 can prove the good performance of BFGS.

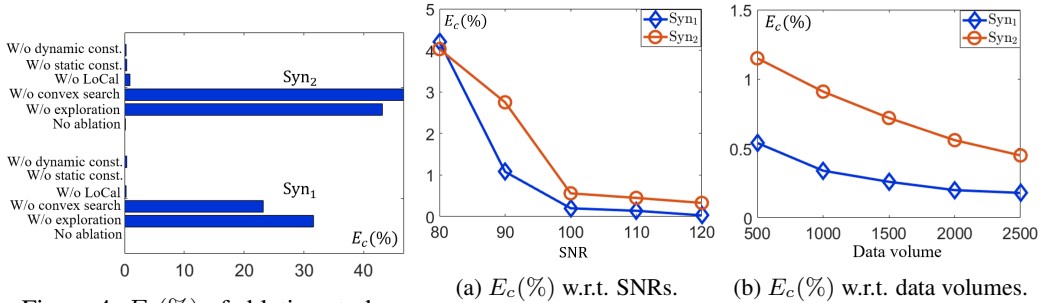

Figure 4: $E_c(\%)$ of ablation study.

(a) $E_c(\%)$ w.r.t. SNRs.  (b) $E_c(\%)$ w.r.t. data volumes.

Figure 5: $E_c(\%)$ of sensitivity analysis.

### 5.5 CONSOLE is Robust with Changing Noise Levels and Data Volume

We utilize $\mathrm{Syn}_1$ and $\mathrm{Syn}_2$ to examine the robustness of the framework with changing noise levels and data volumes. For the noise level, we consider the Signal-to-Noise Ratio (SNR) such that $\mathrm{SNR} \in \{80, 90, 100, 110, 120\}$. For the data volume, we fix $\mathrm{SNR} = 100$ and vary $N \in \{500, 1000, 1500, 2000, 2500\}$. Fig. 5a and 5b demonstrate the results. We find that when $\mathrm{SNR} \geq 100$, the error can be less than $1\%$. This noise level is suitable to real-word systems. For example, $\mathrm{SNR} = 125$ for electric measurements [38]. For the data volume, the overall error is less than $1.5\%$ when $N \geq 500$, which shows a robust performance of CONSOLE.

## 6 Conclusions and Future Work

In this paper, we propose CONSOLE, a novel Convex Neural Symbolic Learning method, which enjoys convexity with certain conditions to tackle Symbolic Regression with guaranteed performances. Specifically, we convexify the search problem by proposing a double convex deep-Q learning. In the meantime, we prove the local and strict convexity of the coefficient estimation in our Locally-Convex equation Learner (LOCAL). To our best knowledge, CONSOLE is the first method that provides guarantees with reasonable assumptions to learn exact equations. Besides, CONSOLE has, at a minimum, a broader impact on the following domains. (1) Convex control for physical systems using Reinforcement Learning. (2) Neural networks in engineering applications with local convexity.

## Acknowledgments and Disclosure of Funding

This work is partially supported by NSF (1947135, 2134079, 1939725, ECCS-1810537, and ECCS-2048288), DARPA (HR001121C0165), ARO (W911NF2110088), DOE (DE-AR00001858-1631 and DE-EE0009355) and AFOSR FA9550-22-1-0294.

