# References

[1] J. V. Beck and K. J. Arnold, *Parameter estimation in engineering and science*. James Beck, 1977.

[2] S.-M. Udrescu and M. Tegmark, "Ai feynman: A physics-inspired method for symbolic regression," *Science Advances*, vol. 6, no. 16, p. eaay2631, 2020.

[3] B. K. Petersen, M. L. Larma, T. N. Mundhenk, C. P. Santiago, S. K. Kim, and J. T. Kim, "Deep symbolic regression: Recovering mathematical expressions from data via risk-seeking policy gradients," in *International Conference on Learning Representations*, 2021. [Online]. Available: https://openreview.net/forum?id=m5Qsh0kBQG

[4] Q. Lu, J. Ren, and Z. Wang, "Using genetic programming with prior formula knowledge to solve symbolic regression problem," *Computational intelligence and neuroscience*, vol. 2016, 2016.

[5] P. Orzechowski, W. La Cava, and J. H. Moore, "Where are we now? a large benchmark study of recent symbolic regression methods," in *Proceedings of the Genetic and Evolutionary Computation Conference*, 2018, pp. 1183–1190.

[6] S. Sahoo, C. Lampert, and G. Martius, "Learning equations for extrapolation and control," in *Proceedings of the 35th International Conference on Machine Learning*, ser. Proceedings of Machine Learning Research, J. Dy and A. Krause, Eds., vol. 80. PMLR, 10–15 Jul 2018, pp. 4442–4450. [Online]. Available: https://proceedings.mlr.press/v80/sahoo18a.html

[7] G. Martius and C. H. Lampert, "Extrapolation and learning equations," *arXiv preprint arXiv:1610.02995*, 2016.

[8] M. Werner, A. Junginger, P. Hennig, and G. Martius, "Informed equation learning," *arXiv preprint arXiv:2105.06331*, 2021.

[9] H. Li and Y. Weng, "Physical equation discovery using physics-consistent neural network (pcnn) under incomplete observability," in *Proceedings of the 27th ACM SIGKDD Conference on Knowledge Discovery & Data Mining*, 2021, pp. 925–933.

[10] Z. Chen, Y. Liu, and H. Sun, "Physics-informed learning of governing equations from scarce data," *Nature communications*, vol. 12, no. 1, pp. 1–13, 2021.

[11] R. Fletcher, *Practical methods of optimization*. John Wiley & Sons, 2013.

[12] T. N. Mundhenk, M. Landajuela, R. Glatt, C. P. Santiago, D. M. Faissol, and B. K. Petersen, "Symbolic regression via neural-guided genetic programming population seeding," *arXiv preprint arXiv:2111.00053*, 2021.

[13] M. L. Larma, B. K. Petersen, S. K. Kim, C. P. Santiago, R. Glatt, T. N. Mundhenk, J. F. Pettit, and D. M. Faissol, "Improving exploration in policy gradient search: Application to symbolic optimization," *arXiv preprint arXiv:2107.09158*, 2021.

[14] J. T. Kim, M. L. Larma, and B. K. Petersen, "Distilling wikipedia mathematical knowledge into neural network models," *arXiv preprint arXiv:2104.05930*, 2021.

[15] L. Biggio, T. Bendinelli, A. Neitz, A. Lucchi, and G. Parascandolo, "Neural symbolic regression that scales," in *International Conference on Machine Learning*. PMLR, 2021, pp. 936–945.

[16] B. K. Petersen, C. Santiago, and M. Landajuela, "Incorporating domain knowledge into neural-guided search via in situ priors and constraints," in *8th ICML Workshop on Automated Machine Learning (AutoML)*, 2021.

[17] B. Amos, L. Xu, and J. Z. Kolter, "Input convex neural networks," in *International Conference on Machine Learning*. PMLR, 2017, pp. 146–155.

[18] C. Rao, P. Ren, Y. Liu, and H. Sun, "Discovering nonlinear pdes from scarce data with physics-encoded learning," *arXiv preprint arXiv:2201.12354*, 2022.

[19] T. Elsken, J. H. Metzen, and F. Hutter, "Neural architecture search: A survey," *The Journal of Machine Learning Research*, vol. 20, no. 1, pp. 1997–2017, 2019.

[20] B. Zoph and Q. V. Le, "Neural architecture search with reinforcement learning," *arXiv preprint arXiv:1611.01578*, 2016.

[21] B. Baker, O. Gupta, N. Naik, and R. Raskar, "Designing neural network architectures using reinforcement learning," *arXiv preprint arXiv:1611.02167*, 2016.

[22] K. O. Stanley, J. Clune, J. Lehman, and R. Miikkulainen, "Designing neural networks through neuroevolution," *Nature Machine Intelligence*, vol. 1, no. 1, pp. 24–35, 2019.

[23] E. Real, A. Aggarwal, Y. Huang, and Q. V. Le, "Regularized evolution for image classifier architecture search," in *Proceedings of the aaai conference on artificial intelligence*, vol. 33, no. 01, 2019, pp. 4780–4789.

[24] C. Liu, B. Zoph, M. Neumann, J. Shlens, W. Hua, L.-J. Li, L. Fei-Fei, A. Yuille, J. Huang, and K. Murphy, "Progressive neural architecture search," in *Proceedings of the European conference on computer vision (ECCV)*, 2018, pp. 19–34.

[25] G. Bender, P.-J. Kindermans, B. Zoph, V. Vasudevan, and Q. Le, "Understanding and simplifying one-shot architecture search," in *International Conference on Machine Learning*. PMLR, 2018, pp. 550–559.

[26] F. Bach, "Breaking the curse of dimensionality with convex neural networks," *The Journal of Machine Learning Research*, vol. 18, no. 1, pp. 629–681, 2017.

[27] M. Pilanci and T. Ergen, "Neural networks are convex regularizers: Exact polynomial-time convex optimization formulations for two-layer networks," in *International Conference on Machine Learning*. PMLR, 2020, pp. 7695–7705.

[28] K. Kawaguchi, "Deep learning without poor local minima," in *Advances in Neural Information Processing Systems*, D. Lee, M. Sugiyama, U. Luxburg, I. Guyon, and R. Garnett, Eds., vol. 29. Curran Associates, Inc., 2016. [Online]. Available: https://proceedings.neurips.cc/paper/2016/file/f2fc990265c712c49d51a18a32b39f0c-Paper.pdf

[29] T. Milne, "Piecewise strong convexity of neural networks," in *Advances in Neural Information Processing Systems*, H. Wallach, H. Larochelle, A. Beygelzimer, F. d'Alché-Buc, E. Fox, and R. Garnett, Eds., vol. 32. Curran Associates, Inc., 2019. [Online]. Available: https://proceedings.neurips.cc/paper/2019/file/b33128cb0089003ddfb5199e1b679652-Paper.pdf

[30] D. Abel, A. Barreto, M. Bowling, W. Dabney, S. Hansen, A. Harutyunyan, M. K. Ho, R. Kumar, M. L. Littman, D. Precup *et al.*, "Expressing non-markov reward to a markov agent."

[31] D. Abel, W. Dabney, A. Harutyunyan, M. K. Ho, M. Littman, D. Precup, and S. Singh, "On the expressivity of markov reward," *Advances in Neural Information Processing Systems*, vol. 34, pp. 7799–7812, 2021.

[32] D. Bertsekas, *Convex optimization algorithms*. Athena Scientific, 2015.

[33] D. P. Kingma and J. Ba, "Adam: A method for stochastic optimization," *arXiv preprint arXiv:1412.6980*, 2014.

[34] T. Hester, M. Vecerik, O. Pietquin, M. Lanctot, T. Schaul, B. Piot, D. Horgan, J. Quan, A. Sendonaris, I. Osband *et al.*, "Deep q-learning from demonstrations," in *Proceedings of the AAAI Conference on Artificial Intelligence*, vol. 32, no. 1, 2018.

[35] V. Mnih, K. Kavukcuoglu, D. Silver, A. A. Rusu, J. Veness, M. G. Bellemare, A. Graves, M. Riedmiller, A. K. Fidjeland, G. Ostrovski *et al.*, "Human-level control through deep reinforcement learning," *nature*, vol. 518, no. 7540, pp. 529–533, 2015.

[36] J. Fan, Z. Wang, Y. Xie, and Z. Yang, "A theoretical analysis of deep q-learning," in *Learning for Dynamics and Control*. PMLR, 2020, pp. 486–489.

[37] J. Yu, Y. Weng, and R. Rajagopal, "Mapping rule estimation for power flow analysis in distribution grids," *arXiv preprint arXiv:1702.07948*, 2017.

[38] H. Li, Y. Weng, Y. Liao, B. Keel, and K. E. Brown, "Distribution grid impedance & topology estimation with limited or no micro-pmus," *International Journal of Electrical Power & Energy Systems*, vol. 129, p. 106794, 2021.

[39] MATPOWER community, "MATPOWER," 2020, https://matpower.org/.


# A  Appendix

## A.1  Model Overview with Pseudo Codes

In this subsection, we provide a high-level summary of our framework for better understanding. We present the summary in the form of pseudo codes, shown in Algorithm 1.

---

**Algorithm 1** The Overview of the Proposed Framework.

---

**Input:** Training dataset $\{\boldsymbol{x}_i, \boldsymbol{y}_i\}_{i=1}^N$.
**Step 1: Design LOCAL.** LOCAL has repeated blocks of symbolic activation, multiplication, and summation layers. For example, Fig. 1 presents a LOCAL with 2 blocks.
**Step 2: Denote LOCAL Function.** LOCAL represents the map $f(\boldsymbol{x}; \{\boldsymbol{Z}_k\}_{k=0}^{K-1}, \{\boldsymbol{W}_k\}_{k=0}^{K-1})$ from $\boldsymbol{x}$ to $\boldsymbol{y}$. With global optimal solutions of $\boldsymbol{Z}_k$ and $\boldsymbol{W}_k$, $f(\boldsymbol{x})$ can be simplified to the true equation $g(\boldsymbol{x})$.
**while** LOCAL does not have the optimal performance **do**
    **Step 3: Search LOCAL Structure.**
    **Step 3.1: Model the Search Process.** Build the CMP and the reward function $R(\cdot)$ based on states and actions defined over LOCAL. Formulate a sequential optimization.
    **Step 3.2: Solve the Optimization.** Utilize the proposed double convex Q-learning to find optimal actions. Generate a search result of $\{\boldsymbol{Z}_k\}_{k=0}^{K-1}$.
    **Step 4: Estimate LOCAL Parameters.** Train the searched LOCAL by minimizing the MSE via Adam. Estimate values in $\{\boldsymbol{W}_k\}_{k=0}^{K-1}$.
    **Step 5: Evaluate the Search and Estimation Results.** The results can formulate $f_t(\boldsymbol{x})$ for the $t^{th}$ episode. Calculate the end-of-trajectory reward $R_t$ to evaluate $f_t(\boldsymbol{x})$.
**Output:** LOCAL with the best performance and the corresponding equations.

---

## A.2  Training Algorithm for CONSOLE.

The training algorithm can be seen in Algorithm 2.

## A.3  Proofs of Theorem 1

**Theorem.** $\forall 0 \leq k \leq K-1$, the negative optimal Q-function $-Q^*(\boldsymbol{s}_k, \tilde{\boldsymbol{a}}_k)$ in the proposed CONSOLE framework exists and is convex in $\boldsymbol{s}_k$ and $\tilde{\boldsymbol{a}}_k$, where $\boldsymbol{s}_k$ is the discrete state and $\tilde{\boldsymbol{a}}_k$ is the continuous action at the $k^{th}$ stage.

*Proof.* First, we show our state transition satisfies the Markov property. Specifically, Equation (1) in our paper shows that the next state $\boldsymbol{s}_{k+1}$ equals the matrix multiplication between the current state $\boldsymbol{s}_k$ and the matrix $\boldsymbol{Z}_k$ that is a matricization of the current action $\boldsymbol{a}_k$, where $k$ is the index of the state. Therefore, the state transition satisfies Markov property with the transition probability $P(\boldsymbol{s}_{k+1}|\boldsymbol{s}_k, \boldsymbol{a}_k) = 1$.

Due to the Markov property of the state transition, we define our search process as Controlled Markov Process (CMP) [31, 30]. By the CMP definition [30], our CMP is composed of our state, action, state transition probability, a discounter factor $\gamma$, and a start state (i.e., $\boldsymbol{s}_0$ in Equation (1)). In general, CMP is a Markov Decision Process (MDP) without a reward function [31].

For one CMP, Trajectory Ordering (TO) ranks trajectories of state action pairs [31]. In our paper, we define the trajectory from $(\boldsymbol{s}_0, \boldsymbol{a}_0)$ to $(\boldsymbol{s}_{K-1}, \boldsymbol{a}_{K-1})$ for $K$-layer LOCAL. Then, our reward function $R(\cdot)$ realizes a TO for our defined trajectories [31] since the ordering of trajectories can be determined by $R(\cdot)$. More specifically, $R(\cdot)$ is trained with the end-of-trajectory reward $R_t$ for the $t^{th}$ trajectory in our paper and can rank trajectories. A reward bundle is an automation-like structure to produce rewards for a CMP [30]. By Corollary 2 of [30], there exists a reward bundle for our defined CMP and TO realized by $R(\cdot)$.

We pair our CMP with the reward bundle to form a Split Partially Observable MDP (Split-POMDP) [30]. Then, by Proposition 1 and Corollary 1 in [30], our Split-POMDP will always have an optimal deterministic policy that only depends on states in our CMP. By the proof of Proposition 1 in [30],

---

**Algorithm 2** CONSOLE: Convex Neural Symbolic Learning

---

**Input:** Training dataset $\{\boldsymbol{x}_i, \boldsymbol{y}_i\}_{i=1}^N$.

**Initialize:** LOCAL layer number $K$, initial state $\boldsymbol{s}_0 = [\mathbf{1}, \mathbf{0}]^T$, discount factor $\gamma \in (0, 1)$, $\epsilon$ for $\epsilon$-greedy strategy, $\lambda$ as a threshold to stop searching, ICNN for reward function $-R(\boldsymbol{s}, \boldsymbol{a})$, ICNN for Q-function $-Q(\boldsymbol{s}, \boldsymbol{a})$, replay buffer $B = \emptyset$, maximum episode $T$, target network $Q'(\cdot) = Q(\cdot)$, and target network update interval $T_0$.

**while** $t \leq T$ **do**
    **while** $k \leq K$ **do**
        Solve Optimization in Equation (3) with $-Q(\boldsymbol{s}_k^t, \boldsymbol{a})$ to obtain $\tilde{\boldsymbol{a}}_k^*$.
        Use $\epsilon$-greedy to select $\tilde{\boldsymbol{a}}_k^t$ from $\tilde{\boldsymbol{a}}_k^*$ and a random action.         ▷ $\epsilon$-greedy strategy.
        Discretize $\tilde{\boldsymbol{a}}_k^t$ to obtain $\boldsymbol{a}_k^t$.
        Execute $\boldsymbol{a}_k^t$ and use Equation (1) to obtain $\boldsymbol{s}_{k+1}^k$.
        Check if $\boldsymbol{a}_k^t$ and $\boldsymbol{s}_{k+1}^k$ satisfy certain constraints. Otherwise, delete this state transition and
restart the iteration from $\boldsymbol{s}_k^t$.         ▷ Constraint checking.
    Formulate LOCAL, train LOCAL with $\{\boldsymbol{x}_i, \boldsymbol{y}_i\}_{i=1}^N$, and calculate $R_t$.
    Train the reward function $-R(\cdot)$ using training data $\{\{\boldsymbol{s}_k^t, \boldsymbol{a}_k^t\}_{k=0}^{K-1}, -R_t\}$.
    $\forall 0 \leq k \leq K$, insert $(\boldsymbol{s}_k^t, \boldsymbol{a}_k^t, \boldsymbol{s}_{k+1}^t, R_t)$ and $(\boldsymbol{s}_k^t, \tilde{\boldsymbol{a}}_k^t, \boldsymbol{s}_{k+1}^t, R(\boldsymbol{s}_k^t, \tilde{\boldsymbol{a}}_k^t))$ to $B_0$.
    Sample a random minibatch $B_0 \subset B$
    **for** $(\boldsymbol{s}_m, \boldsymbol{a}_m, \boldsymbol{s}_{m+1}, R_m) \in B_0$ **do**         ▷ Experience replay.
        Solve Optimization in Equation (3) with $-Q'(\boldsymbol{s}_{m+1}, \boldsymbol{a})$ to obtain $\tilde{\boldsymbol{a}}_{m+1}$.
        $y_m = R_m + \gamma Q'(\boldsymbol{s}_m, \boldsymbol{a}_m)$.
    Train $Q(\cdot)$ using training data $\{\boldsymbol{s}_{m+1}, \boldsymbol{a}_{m+1}, y_m\}_m$, where $\{\boldsymbol{s}_{m+1}, \boldsymbol{a}_{m+1}\}_m$ are the input and $\{y_m\}_m$ are the output.
    **if** $t \mod T_0 = 0$ **then**
        $Q'(\cdot) = Q(\cdot)$         ▷ Update target Q-network.
    **if** $|R_t - 1| \leq \lambda$ **then**
        End the search process.

**Output:** LOCAL with the best performance and the corresponding equations.

---

the optimal policy optimizes the value function over states in CMP. Further, the value function is an evaluation of trajectories for our TO by the proof in Corollary 2 in [30]. Additionally, our TO is realized by our proposed reward function $R(\cdot)$. Therefore, the optimal Q-function exists for our CMP and our proposed $R(\cdot)$.

Then, we consider the Bellman Equation of $Q^*(\cdot)$:

$$-Q^*(\boldsymbol{s}_k, \tilde{\boldsymbol{a}}_k) = -\mathbb{E}[R(\boldsymbol{s}_k, \tilde{\boldsymbol{a}}_k) + \gamma \max_{\boldsymbol{a}} Q^*(\boldsymbol{s}_{k+1}, \tilde{\boldsymbol{a}})] = -R(\boldsymbol{s}_k, \tilde{\boldsymbol{a}}_k) - \gamma \max_{\tilde{\boldsymbol{a}}} Q^*(\boldsymbol{s}_{k+1}, \tilde{\boldsymbol{a}}), \quad (4)$$

where the second equality holds since our state transitions are deterministic by Equation (1). We prove the convexity from the induction method. When $k = K - 1$, the $(k+1)^{th}$ state is the terminal state without action selections. Thus, we have

$$-Q^*(\boldsymbol{s}_{K-1}, \tilde{\boldsymbol{a}}_{K-1}) = -R(\boldsymbol{s}_{K-1}, \tilde{\boldsymbol{a}}_{K-1}).$$

Since $-R(\cdot)$ is an ICNN and is convex in input, $-Q^*(\boldsymbol{s}_{K-1}, \tilde{\boldsymbol{a}}_{K-1})$ is convex in $\boldsymbol{s}_{K-1}$ and $\tilde{\boldsymbol{a}}_{K-1}$.

When $0 \leq k < K - 1$ and assume $-Q^*(\boldsymbol{s}_{k+1}, \tilde{\boldsymbol{a}}_{k+1})$ is convex in $\boldsymbol{s}_{k+1}$ and $\tilde{\boldsymbol{a}}_{k+1}$, we have $-\max_{\tilde{\boldsymbol{a}}} Q^*(\boldsymbol{s}_{k+1}, \tilde{\boldsymbol{a}}) = \min_{\tilde{\boldsymbol{a}}} -Q^*(\boldsymbol{s}_{k+1}, \tilde{\boldsymbol{a}})$ is convex in $\boldsymbol{s}_{k+1}$ given the fixed optimal action. Let $\boldsymbol{H}$ denote the Hessian matrix of $\min_{\tilde{\boldsymbol{a}}} -Q^*(\boldsymbol{s}_{k+1}, \tilde{\boldsymbol{a}})$ with respect to $\boldsymbol{s}_{k+1}$. Due to the convexity, $\boldsymbol{H}$ is positive semi-definite. Thus, by Equation (1) and the chain rule, the Hessian matrix of $\min_{\tilde{\boldsymbol{a}}} -Q^*(\boldsymbol{s}_{k+1}, \tilde{\boldsymbol{a}})$ with respect to $\boldsymbol{s}_k$ can be written as:

$$\boldsymbol{H}' = (\boldsymbol{Z}_k')^T \boldsymbol{H} \boldsymbol{Z}_k'.$$

$\boldsymbol{H}'$ is also positive semi-definite. Therefore, $\min_{\tilde{\boldsymbol{a}}} -Q^*(\boldsymbol{s}_{k+1}, \tilde{\boldsymbol{a}})$ is convex in $\boldsymbol{s}_k$. Since $-R(\boldsymbol{s}_k, \tilde{\boldsymbol{a}}_k)$ is convex in $\boldsymbol{s}_k$, $-Q^*(\boldsymbol{s}_k, \tilde{\boldsymbol{a}}_k)$ is convex in $\boldsymbol{s}_k$.

Similarly, vectorizing the state transition equation can give:

$$\boldsymbol{s}_{k+1} = (\boldsymbol{s}_k^T \bigotimes \boldsymbol{I}_{n_s})\boldsymbol{a}_k^{'},$$

where $\boldsymbol{I}_{n_s}$ is the $n_s \times n_s$ identity matrix and $\bigotimes$ is the Kronecker product. $\boldsymbol{a}_k^{'} = [(\boldsymbol{a}_k)^T, \boldsymbol{0}]^T$ is the concatenation of the discrete action $\boldsymbol{a}_k$ and a zero vector to maintain the fixed dimensionality of action vectors. With similar proofs based on the Hessian matrix and the fact that $-Q^*(\boldsymbol{s}_{k+1}, \tilde{\boldsymbol{a}}_k)$ is convex in $\boldsymbol{s}_{k+1}$, we have $\min_{\tilde{\boldsymbol{a}}} -Q^*(\boldsymbol{s}_{k+1}, \tilde{\boldsymbol{a}})$ is convex in $\boldsymbol{a}_k^{'}$ and also $\boldsymbol{a}_k$. Subsequently, arbitrary $\tilde{\boldsymbol{a}}_k \in \text{conv}(\{0, 1\}^{n_a})$ can be written as a convex combination of the discrete actions $\boldsymbol{a}_k$. Thus, $\min_{\tilde{\boldsymbol{a}}} -Q^*(\boldsymbol{s}_{k+1}, \tilde{\boldsymbol{a}})$ is convex in $\tilde{\boldsymbol{a}}_k$. Since $-R(\boldsymbol{s}_k, \tilde{\boldsymbol{a}}_k)$ is convex in $\tilde{\boldsymbol{a}}_k$, $-Q^*(\boldsymbol{s}_k, \tilde{\boldsymbol{a}}_k)$ is convex in $\tilde{\boldsymbol{a}}_k$. Eventually, $-Q^*(\boldsymbol{s}_k, \tilde{\boldsymbol{a}}_k)$ is convex in $\boldsymbol{s}_k$ and $\tilde{\boldsymbol{a}}_k$, which concludes the proof. ∎

## A.4 Proofs of Theorem 2

**Theorem.** *Let $f^*(\cdot; W)$ denote the* LOCAL *constructed by the optimal sequences of states $(\boldsymbol{s}_0, \boldsymbol{s}_1^*, \cdots, \boldsymbol{s}_K^*)$ and actions $(\boldsymbol{a}_0^*, \boldsymbol{a}_1^*, \cdots, \boldsymbol{a}_{K-1}^*)$ from $-Q^*(\cdot)$, where $W$ is the set of weights of $f^*(\cdot; W)$. If $f^*(\cdot; W)$ can be trained with noiseless datasets and the training can achieve the global optimal weights $W^*$, $f^*(\cdot; W^*)$ can be simplified to the true equation $g(\cdot)$.*

*Proof.* If $f^*(\cdot; W)$ can't represent the exact equations, there are two cases: (1) the structure of $f^*(\cdot; W)$ is correct to represent the equations, but the learned weights $W^*$ don't represent the symbol coefficients, and (2) the structure of $f^*(\cdot; W)$ can't represent the equations. Case (1) doesn't hold since we assume $W^*$ is the global optimal weights for noiseless data. If case (2) holds, $\exists 0 \le j \le K - 1$, $\boldsymbol{b}_j^* = \min_{\tilde{\boldsymbol{a}}_j} -Q^*(\boldsymbol{s}_j, \tilde{\boldsymbol{a}}_j)$ and $\boldsymbol{b}_j^*$ doesn't represent the symbol connections in the underlying equations. Further, we assume $\forall 0 \le i < j$, $\boldsymbol{a}_i^* = \min_{\tilde{\boldsymbol{a}}_i} -Q(\boldsymbol{s}_i, \tilde{\boldsymbol{a}}_i)$ and $\boldsymbol{a}_i^*$ represents the true connections.

If $j = K - 1$, Equation (4) implies that $\tilde{\boldsymbol{a}}_j^* = \min_{\tilde{\boldsymbol{a}}_j} -Q^*(\boldsymbol{s}_j, \tilde{\boldsymbol{a}}_j) = \arg \min_{\tilde{\boldsymbol{a}}} -R(\boldsymbol{s}_j, \tilde{\boldsymbol{a}})$. Since $-R(\boldsymbol{s}_j, \tilde{\boldsymbol{a}})$ is convex in $\tilde{\boldsymbol{a}}$, we know the discrete version of $\tilde{\boldsymbol{a}}_j^*$, namely $\boldsymbol{a}_j^*$, represents the true connection of the last layer for the underlying equations. Otherwise, the reward is not maximized. However, by definition of $\boldsymbol{b}_j^*$, $\boldsymbol{b}_j^* \ne \boldsymbol{a}_j^*$.

If $j < K - 1$, Equation (4) implies:

$$
\begin{aligned}
\min_{\tilde{\boldsymbol{a}}_j} -Q^*(\boldsymbol{s}_j, \tilde{\boldsymbol{a}}_j) = \min_{\tilde{\boldsymbol{a}}_j} -R(\boldsymbol{s}_j, \tilde{\boldsymbol{a}}_j) &+ \gamma \min_{\tilde{\boldsymbol{a}}_j} \min_{\tilde{\boldsymbol{a}}_{j+1}} -R\big(\boldsymbol{s}_{j+1}(\tilde{\boldsymbol{a}}_j), \tilde{\boldsymbol{a}}_{j+1}\big) \\
&+ \cdots + \gamma^{K-1-j} \min_{\tilde{\boldsymbol{a}}_j} \cdots \min_{\tilde{\boldsymbol{a}}_{K-1}} -R(\boldsymbol{s}_{K-1}(\tilde{\boldsymbol{a}}_j, \cdots, \tilde{\boldsymbol{a}}_{K-2}), \tilde{\boldsymbol{a}}_{K-1}).
\end{aligned}
\tag{5}
$$

By definition of $\boldsymbol{b}_j^*$, $\boldsymbol{b}_j^*$ is not the solution of Equation (5). This is because $\boldsymbol{b}_j^*$ can't achieve the minimum value for each summation term on the right hand side of Equation (5), according to the convexity of the reward function. In general, $\boldsymbol{b}_j^* \ne \min_{\tilde{\boldsymbol{a}}_j} -Q^*(\boldsymbol{s}_j, \tilde{\boldsymbol{a}}_j)$, which contradicts the definition of $\boldsymbol{b}_j^*$. Thus, $\boldsymbol{b}_j^*$ doesn't exist. Therefore, case (2) doesn't hold and $f^*(\cdot; W^*)$ represents the exact equations. ∎

## A.5 Proofs of Theorem 3

**Theorem.** *Assume the following conditions hold: (1) the equation $g(\boldsymbol{x})$ is $C^2$ smooth and has bounded second derivatives with respect to weights, (2) $\exists \boldsymbol{x} \in \mathcal{X}$, $g(\boldsymbol{x})$ has non-zero gradients with respect to weights, (3) the structure of* LOCAL *is correctly searched to exactly represent symbols and symbol connections in $g(\boldsymbol{x})$, and (4) the training dataset of* LOCAL *is noiseless. Then, for the MSE loss surface of* LOCAL*, each global optimal point has a strictly convex local region.*

*Proof.* To simplify the proof, we consider scalar output of the LOCAL, i.e., one equation, and the proof can be easily extended to the multi-output case. We follow the idea of [29] to study the second derivative of LOCAL with perturbations. Let $\hat{y}(\boldsymbol{x}, W)$ denote the LOCAL with input to be $\boldsymbol{x}$ and the weight set to be $W$. Let $X$ be a perturbation direction of $W$ and $t$ be a small step size. For the $i^{th}$ noiseless instance $(\boldsymbol{x}_i, y_i)$, we denote $e(\boldsymbol{x}_i, W + tX) = \hat{y}(\boldsymbol{x}_i, W + tX) - y_i$. Obviously, the loss

function can be written as $L(W + tX) = \frac{1}{2N}\sum_{i=1}^{N}(e(\boldsymbol{x}_i, W + tX))^2$. Then, we can calculate the second-order derivative based on the chain rule:

$$\frac{d^2}{dt^2}\Big|_{t=0}L(W + tX) = \frac{1}{N}\frac{d}{dt}\Big|_{t=0}\sum_{i=1}^{N}e(\boldsymbol{x}_i, W + tX)\frac{d}{dt}\hat{y}(\boldsymbol{x}_i, W + tX),$$

$$= \frac{1}{N}\sum_{i=1}^{N}\big(\frac{d}{dt}\big|_{t=0}\hat{y}(\boldsymbol{x}_i, W + tX)\big)^2 + e(\boldsymbol{x}_i, W)\frac{d^2}{dt^2}\Big|_{t=0}\hat{y}(\boldsymbol{x}_i, W + tX). \tag{6}$$

Next, we denote the global optimal solution to be $W^*$. Based on the Assumptions (3) and (4), $\forall i, \hat{y}(\boldsymbol{x}_i, W^*) = g(\boldsymbol{x}_i) = y_i$. Therefore, we have $\frac{d^2}{dt^2}\big|_{t=0}L(W^* + tX) = \frac{1}{N}\sum_{i=1}^{N}\big(\frac{d}{dt}\big|_{t=0}\hat{y}(\boldsymbol{x}_i, W^*)\big)^2 > 0$, where the inequality strictly holds. This is because by Assumptions (3), $\hat{y}(\boldsymbol{x}, W^*)$ can be mathematically simplified to obtain $g(\boldsymbol{x})$. Then, by Assumption (2), $\frac{1}{N}\sum_{i=1}^{N}\big(\frac{d}{dt}\big|_{t=0}\hat{y}(\boldsymbol{x}_i, W^*)\big)^2 > 0$. Finally, by Assumption (1) and (3), $\frac{d^2}{dt^2}\big|_{t=0}\hat{y}(\boldsymbol{x}_i, W + tX)$ is bounded and there is a local region around $W^*$ such that $\frac{d^2}{dt^2}\big|_{t=0}L(W + tX) > 0$, which concludes the proof. ∎

### A.6 Proofs of Theorem 4

**Theorem.** *Suppose Assumptions 1-4 in Theorem 3 hold. For a* LOCAL *with one symbolic activation, multiplication, and summation layer, the set of local convex regions with global optima is* $U = \{W \mid \frac{\left|\frac{d}{dt}\big|_{t=0}\hat{y}(\boldsymbol{x}_i, W + tX)\right|^2}{\eta\left|\frac{d}{dt}\big|_{t=0}\hat{y}(\boldsymbol{x}_j, W + tX)\right|} > |\hat{y}(\boldsymbol{x}_k, W) - y_k|\}$, *where notations are defined in the proof.*

*Proof.* For the target LOCAL, we similarly consider the scalar output and write the function analytically:

$$\hat{y}(\boldsymbol{x}, W) = \boldsymbol{W}_1^T\Psi\big(\Phi(\boldsymbol{W}_0^T\boldsymbol{x})\big), \tag{7}$$

where $\boldsymbol{W}_0 \in \mathbb{R}^{n_0 \times n_1}$ is the weight matrix for activation, $\Phi : \mathbb{R}^{n_1} \to \mathbb{R}^{n_1}$ represents the activation with symbol functions like $x^2$, $cos(x)$, and $log(x)$, etc. $\Psi : \mathbb{R}^{n_1} \to \mathbb{R}^{n_2}$ is the function to select some activated neurons for multiplications, and $\boldsymbol{W}_1 \in \mathbb{R}^{n_2 \times n_3}$ ($n_3 = 1$) represents the weight for summation. We rewrite Equation (7) with the help of exponential and logarithm mappings.

$$\hat{y}(\boldsymbol{x}, W) = \boldsymbol{W}_1^T\exp\Big(\boldsymbol{S}^T\log\big(\Phi(\boldsymbol{W}_0^T\boldsymbol{x})\big)\Big), \tag{8}$$

where $\boldsymbol{S} \in \mathbb{R}^{n_1 \times n_2}$ represents a selection matrix such that $\boldsymbol{S}[i, j] = 1$ if and only if the $i^{th}$ neuron is selected as the multiplicative factor for the $j^{th}$ neuron in the multiplication layer. Given the fixed structure of $\hat{y}(\cdot)$ from the deep Q-learning, $\boldsymbol{S}$ is a known matrix. $\log(\cdot)$ and $\exp(\cdot)$ represent the element-wise logarithm and exponential functions. Notably, the corresponding element in $\Phi(\boldsymbol{W}_0^T\boldsymbol{x})$ should be positive in Equation (8). If there are negative entries, one can utilize $\boldsymbol{W}_1^T\boldsymbol{s} \circ \exp\Big(\boldsymbol{S}^T\log\big(|\Phi(\boldsymbol{W}_0^T\boldsymbol{x})|\big)\Big)$ to take place of the right hand side term in Equation (8), where $\boldsymbol{s}[i] = (-1)^{n_-^i}$ and $0 \le n_-^i \le n_1$ represents the number of negative entries selected for the $i^{th}$ neuron of the multiplication layer. $\circ$ represents the Hadamard product. However, both expressions have the same values and gradients. Thus, we utilize Equation (8) in later derivations.

Then, let $X$ be a perturbation direction such that $X = \{\boldsymbol{X}_0, \boldsymbol{X}_1\}$. Thus, for a small step $t$, we have:

$$\hat{y}(\boldsymbol{x}, W + tX) = (\boldsymbol{W}_1 + t\boldsymbol{X}_1)^T\exp\Big(\boldsymbol{S}^T\log\big(\Phi((\boldsymbol{W}_0 + t\boldsymbol{X}_0)^T\boldsymbol{x})\big)\Big). \tag{9}$$

Based on Equation (9), we can compute:

$$\frac{d}{dt}\hat{y}(\boldsymbol{x}_i, W + tX) = \boldsymbol{X}_1^T \exp\left(\boldsymbol{S}^T \log\big(\Phi((\boldsymbol{W}_0 + t\boldsymbol{X}_0)^T \boldsymbol{x}_i)\big)\right)$$

$$+ (\boldsymbol{W}_1 + t\boldsymbol{X}_1)^T \bigg[\exp\left(\boldsymbol{S}^T \log\big(\Phi((\boldsymbol{W}_0 + t\boldsymbol{X}_0)^T \boldsymbol{x}_i)\big)\right) \qquad (10)$$

$$\circ \boldsymbol{S}^T \frac{1}{\Phi((\boldsymbol{W}_0 + t\boldsymbol{X}_0)^T \boldsymbol{x}_i)} \circ \Phi^{'}((\boldsymbol{W}_0 + t\boldsymbol{X}_0)^T \boldsymbol{x}_i) \circ \boldsymbol{X}_0^T \boldsymbol{x}_i\bigg],$$

where $\frac{1}{\Phi((\boldsymbol{W}_0 + t\boldsymbol{X}_0)^T \boldsymbol{x}_i)} \in \mathbb{R}^{n_1}$ is the element-wise division and $\Phi^{'}$ is the element-wise first derivative of $\Phi^{'}$. Without special notifications, we assume all the division for vectors is element-wise in the following derivations. Then, we denote

$$\boldsymbol{u}(\boldsymbol{x}_i, W + tX) = \exp\left(\boldsymbol{S}^T \log\big(\Phi((\boldsymbol{W}_0 + t\boldsymbol{X}_0)^T \boldsymbol{x}_i)\big)\right),$$

$$\boldsymbol{v}(\boldsymbol{x}_i, W + tX) = \boldsymbol{S}^T \frac{1}{\Phi((\boldsymbol{W}_0 + t\boldsymbol{X}_0)^T \boldsymbol{x}_i)} \circ \Phi^{'}((\boldsymbol{W}_0 + t\boldsymbol{X}_0)^T \boldsymbol{x}_i) \circ \boldsymbol{X}_0^T \boldsymbol{x}_i, \qquad (11)$$

$$\boldsymbol{w}(\boldsymbol{x}_i, W + tX) = \boldsymbol{S}^T \frac{1}{\Phi^{'}((\boldsymbol{W}_0 + t\boldsymbol{X}_0)^T \boldsymbol{x}_i)} \circ \Phi^{''}((\boldsymbol{W}_0 + t\boldsymbol{X}_0)^T \boldsymbol{x}_i) \circ \boldsymbol{X}_0^T \boldsymbol{x}_i.$$

With above definitions, we can calculate:

$$\frac{d}{dt}\Big|_{t=0}\hat{y}(\boldsymbol{x}_i, W + tX) = \boldsymbol{X}_1^T \boldsymbol{u}(\boldsymbol{x}_i, W) + \boldsymbol{W}_1^T \big[\boldsymbol{u}(\boldsymbol{x}_i, W) \circ \boldsymbol{v}(\boldsymbol{x}_i, W)\big]. \qquad (12)$$

Further, we calculate the second derivative based on Equation (10) and the fact that element-wise operations for vectors are commutative:

$$\frac{d}{dt^2}\hat{y}(\boldsymbol{x}_i, W + tX) = \boldsymbol{X}_1^T \big[\boldsymbol{u}(\boldsymbol{x}_i, W + tX) \circ \boldsymbol{v}(\boldsymbol{x}_i, W + tX)\big]$$

$$+ \boldsymbol{X}_1^T \big[\boldsymbol{u}(\boldsymbol{x}_i, W + tX) \circ \boldsymbol{v}(\boldsymbol{x}_i, W + tX)\big]$$

$$+ (\boldsymbol{W}_1 + t\boldsymbol{X}_1)^T \big[\boldsymbol{u}(\boldsymbol{x}_i, W + tX) \circ \boldsymbol{v}(\boldsymbol{x}_i, W + tX) \circ \boldsymbol{v}(\boldsymbol{x}_i, W + tX)\big] \qquad (13)$$

$$- (\boldsymbol{W}_1 + t\boldsymbol{X}_1)^T \big[\boldsymbol{u}(\boldsymbol{x}_i, W + tX) \circ \boldsymbol{v}(\boldsymbol{x}_i, W + tX) \circ \boldsymbol{v}(\boldsymbol{x}_i, W + tX)\big]$$

$$+ (\boldsymbol{W}_1 + t\boldsymbol{X}_1)^T \big[\boldsymbol{u}(\boldsymbol{x}_i, W + tX) \circ \boldsymbol{v}(\boldsymbol{x}_i, W + tX) \circ \boldsymbol{w}(\boldsymbol{x}_i, W + tX)\big],$$

When $t \to 0$, we have:

$$\frac{d}{dt^2}\Big|_{t=0}\hat{y}(\boldsymbol{x}_i, W + tX) = 2\boldsymbol{X}_1^T \big[\boldsymbol{u}(\boldsymbol{x}_i, W) \circ \boldsymbol{v}(\boldsymbol{x}_i, W)\big] + \boldsymbol{W}_1^T \big[\boldsymbol{u}(\boldsymbol{x}_i, W) \circ \boldsymbol{v}(\boldsymbol{x}_i, W) \circ \boldsymbol{w}(\boldsymbol{x}_i, W)\big]$$
$$(14)$$

The above equation can reflect the relationship between the second and the first derivative. However, we first identify the inequality between these two derivatives to enable a strictly convex region.

Let $\hat{\boldsymbol{y}}^{'} = [\frac{d}{dt}\big|_{t=0}\hat{y}(\boldsymbol{x}_1, W + tX), \cdots, \frac{d}{dt}\big|_{t=0}\hat{y}(\boldsymbol{x}_N, W + tX)]^T$, $\hat{\boldsymbol{y}}^{''} = [\frac{d}{dt^2}\big|_{t=0}\hat{y}(\boldsymbol{x}_1, W + tX), \cdots, \frac{d}{dt^2}\big|_{t=0}\hat{y}(\boldsymbol{x}_N, W + tX)]^T$, and $\boldsymbol{e} = [e(\boldsymbol{x}_1, W), \cdots, e(\boldsymbol{x}_N, W)]^T$. Equation (6) implies that:

$$\frac{d^2}{dt^2}\Big|_{t=0}L(W + tX) = \frac{1}{N}(\|\hat{\boldsymbol{y}}^{'}\|_2^2 + \boldsymbol{e}^T \hat{\boldsymbol{y}}^{''})$$

$$\geq \frac{1}{N}(\|\hat{\boldsymbol{y}}^{'}\|_2^2 - \|\boldsymbol{e}\|_2\|\hat{\boldsymbol{y}}^{''}\|_2) \qquad (15)$$

To find a region to restrict the convexity, we restrict the lower bound of the second derivative to be positive and compute:

$$\|\boldsymbol{e}\|_2 < \frac{\|\hat{\boldsymbol{y}}^{'}\|_2^2}{\|\hat{\boldsymbol{y}}^{''}\|_2} \qquad (16)$$

The right hand side of Equation (16) can be easily bounded by:

$$\frac{||\hat{\boldsymbol{y}}^{'}||_2^2}{||\hat{\boldsymbol{y}}^{''}||_2} \geq \frac{\sqrt{N}\min(|\hat{\boldsymbol{y}}^{'}|)^2}{\max(|\hat{\boldsymbol{y}}^{''}|)} = \frac{\sqrt{N}\left|\frac{d}{dt}\big|_{t=0}\hat{y}(\boldsymbol{x}_i, W+tX)\right|^2}{\left|\frac{d}{dt^2}\big|_{t=0}\hat{y}(\boldsymbol{x}_j, W+tX)\right|}, \tag{17}$$

where $|\cdot|$ for a vector is to calculate the absolute value for each element of the vector, $i = \arg\min(|\hat{\boldsymbol{y}}^{'}|)$ and $j = \arg\max(|\hat{\boldsymbol{y}}^{''}|)$. Namely, we consider a sufficient condition for convexity.

$$\frac{\sqrt{N}\left|\frac{d}{dt}\big|_{t=0}\hat{y}(\boldsymbol{x}_i, W+tX)\right|^2}{\left|\frac{d}{dt^2}\big|_{t=0}\hat{y}(\boldsymbol{x}_j, W+tX)\right|} > ||\boldsymbol{e}||_2 \tag{18}$$

Next, Equation (14) indicates that:

$$\left|\frac{d}{dt^2}\big|_{t=0}\hat{y}(\boldsymbol{x}_j, W+tX)\right| = \left|\boldsymbol{X}_1^T\big[\boldsymbol{u}(\boldsymbol{x}_j, W) \circ 2\boldsymbol{v}(\boldsymbol{x}_j, W)\big] + \boldsymbol{W}_1^T\big[\boldsymbol{u}(\boldsymbol{x}_j, W) \circ \boldsymbol{v}(\boldsymbol{x}_j, W) \circ \boldsymbol{w}(\boldsymbol{x}_j, W)\big]\right|$$

$$\leq \eta\Big(\left|\boldsymbol{X}_1^T\boldsymbol{u}(\boldsymbol{x}_j, W) + \boldsymbol{W}_1^T\big[\boldsymbol{u}(\boldsymbol{x}_j, W) \circ \boldsymbol{v}(\boldsymbol{x}_j, W)\big]\right|\Big)$$

$$= \eta\left|\frac{d}{dt}\big|_{t=0}\hat{y}(\boldsymbol{x}_j, W+tX)\right|, \tag{19}$$

where $\eta$ is a positive constant. Note that $\eta < \infty$ by Assumptions (1) and (2) in Theorem 3. Therefore, we have the following sufficient condition to make $\frac{d^2}{dt^2}\big|_{t=0}L(W+tX) > 0$ always hold.

$$\frac{\sqrt{N}\left|\frac{d}{dt}\big|_{t=0}\hat{y}(\boldsymbol{x}_i, W+tX)\right|^2}{\eta\left|\frac{d}{dt}\big|_{t=0}\hat{y}(\boldsymbol{x}_j, W+tX)\right|} > \sqrt{N}|\hat{y}(\boldsymbol{x}_k, W) - y_k| \geq ||\boldsymbol{e}||_2, \tag{20}$$

where $k = \arg\max(|\boldsymbol{e}|)$. The above equation leads to a set $U$ of local regions that have strong convexity. Namely,

$$U = \{W\big| \frac{\left|\frac{d}{dt}\big|_{t=0}\hat{y}(\boldsymbol{x}_i, W+tX)\right|^2}{\eta\left|\frac{d}{dt}\big|_{t=0}\hat{y}(\boldsymbol{x}_j, W+tX)\right|} > |\hat{y}(\boldsymbol{x}_k, W) - y_k|\}. \tag{21}$$

Clearly, the global optimal solution $W^* \in U$ since $\hat{y}(\boldsymbol{x}_k, W^*) - y_k = 0$. Note that there may be multiple global optimal solutions of the loss minimization in LOCAL. Thus, $U$ is the set of local convex regions that contain global optima. This implies that for each $W^* \in U$, we can find a locally and strictly convex region $U^* = U \cap B(r)$, where $B(r) = ||\boldsymbol{w} - \boldsymbol{w}^*||_2 \leq r$ is a norm ball and we vectorize $W$ and $W^*$ to obtain $\boldsymbol{w}$ and $\boldsymbol{w}^*$, respectively. Subsequently, range $r$ can be set relatively large such that $U^* \subset B(r)$ and $U^{**} \cap B(r) = \emptyset$, where $U^{**}$ is the local region for another global optimal point $W^{**}$ if it exists. Then, the range for $U^*$ still depends on the inequality in Equation (21). ∎

## A.7 Implementing details of CONSOLE

Hyper-parameters of CONSOLE exist for both the double convex deep Q-learning and the LOCAL. In the deep Q-learning, we set $\gamma = 0.2$, $\epsilon = 0.4$, $T = 600$, $\lambda = 10^{-2}$, $T_0 = 10$ for Algorithm 2. Furthermore, to train the negative Q-function and the reward function, we set the learning rate to be $5 \times 10^{-3}$ and the number of epochs for training to be 50. Then, we set the batch size for the negative Q-function to be 100. If the number of data in the replay buffer is less than 100, no training happens for the negative Q-function. Additionally, all the data gathered in one episode are used to train the negative reward function. As for the LOCAL, we set $K = 3$, the learning rate to be $1 \times 10^{-2}$ and the number of training epochs to be 8. We make these training epochs to be small since training the LOCAL is the most time-consuming part of CONSOLE. Furthermore, if the structure of LOCAL is correctly searched, a small number of iterations can help LOCAL to gain the global optimal weights. Finally, we initialize all trainable weights in LOCAL to be 1. The following results show that a relatively large area is suitable for an initial guess of LOCAL.