# OpenReview forum: "CoNSoLe: Convex Neural Symbolic Learning"
_NeurIPS.cc/2022/Conference — NeurIPS 2022 Accept_

### Official Review · Reviewer_EjER · 2022-07-09

**Rating:** 7
**Confidence:** 3
**Soundness:** 4 excellent
**Presentation:** 3 good
**Contribution:** 4 excellent

**Summary:**

The authors propose a method for symbolic regression based on deep Q-learning, that learns to chose which symbols to combine in the LoCal model. To provide formal guarantees, the authors use input convex neural networks to model both the Q-function and the Reward function. Experiments suggest that the method is able to accurately recover true underlying equations from data.

**Questions:**

Initially I was puzzled as to why the authors were so keen on maintaining convexity in R and Q for the deep Q-learning procedure. Is it just such the optimal action is uniquely defined and there's an algorithm to find it? Or does it help with the large state and action space?

When R is convex, the optimal Q is convex. But how reasonable is it to model R as a convex function?


**Limitations:**

The authors have adequately addressed the limitations of their work.

**Strengths And Weaknesses:**

Strength

- Elegant use of input convex neural networks for Deep Q-learning
- Theorem 1 shows that modeling Q as a convex function is a consequence of modeling R as a convex function
- Theorem 3 shows that the loss landscape of chosing optimal multiplicative weights in the model has local convex region under mild assumptions. Hence we can find it with gradient descent methods if the initialization is lucky.
- Strong experimental evaluation on synthetic problems as well as systems from physics and electrical engineering. Several benchmark methods are considered, the ableation study highlights the importance of each component of the contribution
- Figure 1 is very informative.

Weaknesses:

-The authors claim  that "Q-learning can hardly be applicable when the state and action spaces are large, e.g., SR problem." but then proceed to do so anyways?
- Code is not available to reviewers, so I cannot confidently attest to reproducibility

Minor Comments

- There are some redundancies between Introduction and Related Work.
- Table 1 violates the margins
- Authors use references [x] as subject in sentences. This should generally be avoided, use author names as subject instead.
- Personally I'd like to have Algorithm~1 in the main text. I know space is tight, but the code helps a lot for understanding the interplay of all the components you propose

---

> ### Author Response · Authors · 2022-08-02
> **Response to Reviewer EjER**
>
> We summarize and answer the questions from the reviewer.
>
> $\textbf{Q1: Vague statement ''Q-learning can hardly be applicable when the state and action spaces are large, e.g., SR problem.''.}$
>
> Answer: Thank you for pointing out this. We have changed the statement from Q-learning to tabular Q-learning in the above sentence.
>
> $\textbf{Q2: Redundancies between Introduction and Related Work.}$
>
> Answer: We removed the redundancy. For example, in Related Work, I have deleted all the literature reviews that have been done in the Introduction. Please refer to the modified paper.
>
> $\textbf{Q3: Table margin, reference style, and presentation of Algorithm 1.}$
>
> Answer: Thanks a lot for your suggestions. We will follow what you suggested to do in the final version of the paper.
>
> $\textbf{Q4: Convexity in $R(\cdot)$ and $Q(\cdot)$ for optimal solutions and large search space.}$
>
> Answer: We think the convexity can benefit both the guarantee and the search for the large search space. First, we prove in our Theorem 1 and 2 that the convexified search process can generate a global optimal action sequence from $-Q^{*}(\cdot)$, which finally results in obtaining the true underlying equation under some moderate assumptions. However, vanilla deep Q learning or Recurrent Neural Network (RNN)-based Reinforcement Learning (RL) do not have this guarantee. Second, we admit that compared to our model, vanilla deep Q learning has a lower computational cost in each iteration [1]. However, the non-convexity in vanilla deep Q learning causes no guarantees to find the true equation, especially when the state and action spaces are large. We have shown the comparison in Fig. 4 in the ablation study. When the size of the search space increases (i.e., from $\text{Syn}_1$ to $\text{Syn}_2$), the error of using vanilla deep Q learning increases by around $25\\%$. However, for our model, the error increases by around $2\\%$.
>
> $\textbf{Q5: Motivation for convexity in $R(\cdot)$.}$
>
> Answer:  Convexification of $R(\cdot)$ and $-Q\^{\*}(\cdot)$ (by our Theorem 1) enjoys two benefits. First, although we have convexified the negative Q function $-Q(\cdot)$, $Q(\cdot)$ may not necessarily converge to the $Q\^{\*}(\cdot)$. What's more, the restriction of the convex shape for $-Q(\cdot)$ may hurt its convergence to $-Q\^{\*}(\cdot)$. Therefore, our extra convexification of $-R(\cdot)$ and $-Q\^{\*}(\cdot)$ guarantees that the convexity of $-Q(\cdot)$ will not hurt the convergence. This benefit is stated in lines 188-192 in the first version of the paper. The second benefit is that the convex $-Q\^{\*}(\cdot)$ can lead to the finding of the true equation by our Theorem 2.
>
> [1] Mnih, Volodymyr, et al. "Playing atari with deep reinforcement learning." 2013.

---

### Official Review · Reviewer_faoT · 2022-07-11

**Rating:** 6
**Confidence:** 2
**Soundness:** 3 good
**Presentation:** 3 good
**Contribution:** 3 good

**Summary:**

The authors propose a two-stage symbolic learning approach, where each stage is solvable with convex optimization. They then provide experiments validating their proposed approach.

**Questions:**

N/A

**Limitations:**

Yes

**Strengths And Weaknesses:**

Strengths
- The use of an ICNN for modeling the negative reward function is quite clever.
- The experimental evaluation seems quite robust. The results compared to competing methods seem very strong. Note that this is not exactly my field, so I am not completely aware if these are good benchmarks to evaluate against.
- The ablations on SNR and different components of CoNSoLe are great to see, especially because the theoretical results only analyze the noiseless case.

Weaknesses
- In the notations section (line 114) it is stated that $h_K = y$, which is not exactly correct. $h_K$ *regresses* or is in some way a *prediction* of $y$, but to say that the output of the network exactly is equal to the labels is not quite precise.
- For global optima in neural networks, some background is lacking. For example, [1] was the first to demonstrate that two-layer ReLU neural networks can actually be expressed as convex optimization problems (over the weights), and thus can be solved to their global optimum.

[1] "Neural networks are convex regularizers", Pilanci & Ergen, 2020

---

> ### Author Response · Authors · 2022-08-02
> **Response to Reviewer faoT**
>
> We summarize and answer the questions from the reviewer.
>
> $\textbf{Q1: Notation typos.}$
>
> Answer: Thanks a lot for pointing out his typo. We change the equation to $\boldsymbol{h}_K=\hat{\boldsymbol{y}}$, where $\hat{\boldsymbol{y}}$ is the output of our Locally Convex Equation Learner (LoCaL).
>
> $\textbf{Q2: Some background is lacking.}$
>
> Answer: Thanks for your information. We add the following descriptions to the related work. Specifically, [1] investigates a single hidden layer with unbounded neurons and non-Euclidean regularization. The authors show the training can be done via convex optimizations. [2] considers finite neurons and develops a novel duality theory to train two-layer NNs as convex optimization problems.
>
> [1] Bach, Francis. "Breaking the curse of dimensionality with convex neural networks." 2017.
>
> [2] Pilanci, Mert, and Tolga Ergen. "Neural networks are convex regularizers: Exact polynomial-time convex optimization formulations for two-layer networks." 2020.

---

### Official Review · Reviewer_ipq6 · 2022-07-12

**Rating:** 6
**Confidence:** 2
**Soundness:** 4 excellent
**Presentation:** 3 good
**Contribution:** 3 good

**Summary:**

This paper considers the problem of symbolic learning using neural networks. The paper is focused on designing symbolic search algorithms with correctness guarantees. This is achieved by using a $Q$-learning framework, where the $Q$ function is modeled using an input convex neural network, ensuring exact temporal difference updates can be computed efficiently. The authors provide theoretical guarantees for the correctness of their algorithm, modulo a requirement to be able to fix neural network parameters exactly (a non-convex problem). This is balanced, however, by a follow up result showing that globally optimal points always lie in (strictly) convex neighborhoods. The method is tested on equation learning tasks, performing favorably.

**Questions:**

- Why are you allowed to restrict the negative reward function to be convex? Of course it is permissible to model the $Q$ function, but rewards usually come from a ground truth.
- Is  $Q(s, a)$ differentiable in $a$? It appears that for non-integral $a$, $Q$ is computed using a thresholding at $0.5$. This seems to me to introduce discontinuities. If so, why does this not make optimizing $\min_a Q(s,a)$ difficult?

**Limitations:**

Yes.

**Strengths And Weaknesses:**

Thanks to the authors for an interesting work on the salient topic of neural-symbolic learning. I enjoyed reading the paper, and certainly felt like I learned something. Some

- The convexity results are really nice, and well supported by the ablations. It is worth noting of course that the overall procedure is not fully convex (which it cannot be due to NP-hardness results) but the non-convexity comes from needing to fit  neural network parameter, arguably a fairly benign form of non-convexity.
- Experiments cover a number of different tasks, and appear thorough. Results seem promising, for instance the proposed method is considerably closer to the ground truth equations in Table 1 than the other methods.

Stepping back a moment, the paper is well organized, although I found it quite information dense. The paper went through the important steps to make the work properly complete, including good, simple theoretical results, and an experiment section with comparisons to alternate methods, and ablations. The ablation section provides useful insights. For instance, the double-convexity of the $Q$ function is doing a lot of heavy lifting—removing it significantly impacts performance. This is encouraging, suggesting that the input convex network design is working well.

In all, this paper presents a method for neural-symbolic learning that appears non-trivial and effective. Importantly, it is accompanied by correctness guarantees for (large parts of) the learning algorithm. This is an especially promising sign in the context of symbolic learning, where finding exact solutions is of comparatively greater importance than in other areas. Overall I am broadly positive about this work. A disclaimer, however, is that I am not familiar with recent related works in this area. This is reason for the given confidence score, and will pay careful attention to any related works that may emerge during discussions.

---

> ### Author Response · Authors · 2022-08-02
> **Response to Reviewer ipq6**
>
> We summarize and answer the questions from the reviewer.
>
> $\textbf{Q1: Why do you restrict $-R(\cdot)$ to be convex?}$
>
> Answer: According to the Bellman Equation (Equation (4) in Appendix A.2), the optimal Q-function $Q\^{\*}(\cdot)$ depends on the reward function $R(\cdot)$ for each state-action pair. Thus, we propose to convexify the $-R(\cdot)$  and prove that this leads to the convexity of the $-Q\^{\*}(\cdot)$ in Theorem 1 (see proofs in Appendix A.2). This procedure enjoys two benefits. First, although we have convexified the negative Q function $-Q(\cdot)$, $Q(\cdot)$ may not necessarily converge to the $Q\^{\*}(\cdot)$. What's more, the restriction of the convex shape for $-Q(\cdot)$ may hurt its convergence to $-Q\^{\*}(\cdot)$. Therefore, our extra convexification of $-R(\cdot)$ and $-Q\^{\*}(\cdot)$ guarantees that the convexity of $-Q(\cdot)$ will not hurt the convergence. This benefit is stated in lines 188-192 in the first version of the paper. The second benefit is that the convex $-Q\^{\*}(\cdot)$ can lead to the finding of the true equation by our Theorem 2.
>
> $\textbf{Q2: Differentiability and difficulty in optimizing $Q(\boldsymbol{s},\boldsymbol{a})$.}$
>
> Answer: $Q(\boldsymbol{s},\boldsymbol{a})$ is not differentiable in $\boldsymbol{a}$ as in our paper, the action $\boldsymbol{a}\in\\{0,1\\}\^{n_a}$ where $n_a$ is the dimension of $\boldsymbol{a}$. Therefore, we introduce a continuous action space $\text{conv}(\\{0,1\\}\^{n_a})$ in line 165 in the first version of the paper, where $\text{conv}(\cdot)$ is the operation to obtain the convex hull. Thus, the convex optimization lies in the continuous action space, as shown in Equation (3) in the paper, where we utilize $\tilde{\boldsymbol{a}}$ to represent the continuous action vector. The continuous optimization for the convex function $-Q(\boldsymbol{s},\tilde{\boldsymbol{a}})$ can be easily solved via the proposed bundle entry method in the input convex neural network paper [1], as shown in line 167. Then, we utilize $0.5$ as a threshold for discretizing the hypercube $\text{conv}(\cdot)$ and convert from $\tilde{\boldsymbol{a}}^*$ to $\boldsymbol{a}^*$, where $\tilde{\boldsymbol{a}}^*$ is the optimal solution for the optimization in Equation (3). $\boldsymbol{a}^*$ is the discrete version and is used to construct our proposed neural network, i.e., Locally Convex Equation Learner (LoCaL).
>
>
> [1] Amos, Brandon, Lei Xu, and J. Zico Kolter. "Input convex neural networks." 2017.

---

> > ### Comment · Reviewer_ipq6 · 2022-08-08
> > **Rebuttal response**
> >
> > Thank you for the time taken to respond to myself and the other reviewers, all of which I have read carefully.
> >
> > The explanation for why $R$ is taken to be convex makes sense from the point of view of "what this buys us" (provable guarantees on finding optima of $Q^*$) but I am not clear on _why it is allowed_ to take $R$ convex---is the reward function something the user gets to define? How do we know there is a convex reward function that adequately describes a given task we may be interested in?
> >
> > In all, I found it very difficult to grasp the main ideas in this paper. This is no doubt in part due to my non-familiarity with certain related literature. However, I am also convinced that the paper has been written in a very dense way that would make it hard for any reader to follow. One simple example comes from the value function $Q$ defined for non-integral $a$: I am still not sure how $Q(s,a)$ is defined from non-integral $a$ as it is never explained. I would kindly encourage the authors to do a thorough re-evaluation of their own exposition to look for ways to make the core ideas more digestible. Doing so will help make this work, which I do believe contains some nice provable methods for considering symbolic problems, more widely read.
> >
> > Best wishes.

---

> > > ### Author Response · Authors · 2022-08-09
> > > **Reply to Reviewer ipq6, 2nd round, part 1**
> > >
> > > Thanks a lot for your insightful questions and useful suggestions. We first show that it's doable to utilize convex and piece-wise linear functions to make $R$ convex. Then, we present the relationship between $Q(\cdot)$ and discrete/continuous actions. Thirdly, we re-organize the logic and presentation of the paper to make readers easily grasp our main ideas. Please see the revised paper with modifications marked in blue. Please note we will further refresh the paper in terms of the margins and spaces in the final submission.
> > >
> > > $\textbf{Q1: Feasibility to let a convex function fit the reward.}$
> > >
> > > Answer: Thanks a lot for this insightful question. Our answer is that a convex reward function is adequate for our task. Essentially, we need our negative reward function $-R(\boldsymbol{s},\boldsymbol{a})$ to $(i)$ maintain a convex shape over the continuous action space $\text{conv}(\\{0,1\\}\^{n\_a})$ and $(ii)$ fit the discrete states and actions with the end-of-trajectory reward, as shown in lines 162-170 in the revised paper. Further, the discrete actions are the endpoints of the hypercube $\text{conv}(\\{0,1\\}\^{n\_a})$ , as shown in lines 174-175. Among these endpoints, the optimal action has the strictly optimal end-of-trajectory reward values. This is because only the optimal action sequences can obtain the highest reward. Thus, we require a convex function $-R(\boldsymbol{s},\boldsymbol{a})$ to fit endpoints in a hypercube with one optimal endpoint to have the lowest value.
> > >
> > > This requirement can be easily achieved via piece-wise linear functions. For example, for $1$-dimensional actions ($n_a=1$), a line can be a convex function to fit the two endpoints. For $2$-dimensional actions, we can utilize planes to fit arbitrary three endpoints. The intersections of these planes can form a convex function with one endpoint to have the lowest value.  Such an idea can be extended to hypercube and hyperplanes, and we can always find a convex function $-R(\boldsymbol{s},\boldsymbol{a})$ to fit the data. The specific fitting process is achieved via training the ICNN of $-R(\boldsymbol{s},\boldsymbol{a})$, as shown in lines 169-170. The ReLu activations in $-R(\boldsymbol{s},\boldsymbol{a})$ can guarantee the piece-wise linearity. For example, in Fig. 2, we visualize $-Q\^{\*}(\boldsymbol{s},\boldsymbol{a})$ which is an expectation of $-R(\boldsymbol{s},\boldsymbol{a})$. The visualization shows that ReLu helps to construct convex and piece-wise linear functions. We illustrate the above statements in lines 175-180.
> > >
> > > $\textbf{Q2: How $Q(\cdot)$ is defined for non-integral action.}$
> > >
> > > Answer: We first define the discrete action $\boldsymbol{a}\in\\{0,1\\}\^{n\_a}$ that are the endpoints of hypercubes in the $n_a$-dimensional space. Then, the convex hull of these endpoints $\text{conv}(\\{0,1\\}\^{n\_a})$, i.e., the hypercube, is a convex and continuous action space. Therefore, the negative Q function $-Q(\boldsymbol{s},\tilde{\boldsymbol{a}})$ is defined over the continuous and convex hypercube, where $\tilde{\boldsymbol{a}}\in\text{conv}(\\{0,1\\}\^{n\_a})$ is a continuous action. Thus, the operations are always conducted over $\text{conv}(\\{0,1\\}\^{n\_a})$. The definitions can be seen in lines 171-175. Then, we have the following operations.
> > >
> > > (1) $\textbf{Initialization}.$ The value function $-Q(\cdot)$ and the reward function $-R(\cdot)$ are randomly initialized via initializing the weights of ICNNs. Then, $Q(\boldsymbol{s},\tilde{\boldsymbol{a}})$ and $R(\boldsymbol{s},\tilde{\boldsymbol{a}})$ can always output values in the continuous action space. It should be noted that $-Q(\cdot)$ and $-R(\cdot)$ are always convex despite the weight values due to the property of ICNN.
> > >
> > > (2) $\textbf{Updating}.$ The updating of the reward function $R(\cdot)$ is to train the ICNN with end-of-trajectory rewards. This makes the endpoints in $\text{Conv}(\\{0,1\\}\^{n\_a})$ to have output values very close to the obtained rewards. The updating of the Q-function $Q(\cdot)$ is through Equation (2) in the paper. Especially, the updating process with optimization in Equation (2) is over $\text{Conv}(\\{0,1\\}\^{n\_a})$ for continuous and convex optimization.
> > >
> > > (3) $\textbf{Discretization}.$  After solving the optimization over the continuous action space, we finally need discrete actions to formulate our LoCaL. Thus, we propose to discretize the continuous optimal action $\tilde{\boldsymbol{a}}\^{\*}\in \text{Conv}(\\{0,1\\}\^{n\_a})$ with a fixed threshold and obtain $\boldsymbol{a}\^{\*}\in\\{0,1\\}\^{n\_a}$. Although $\boldsymbol{a}^{\*}$ is usually sub-optimal for $-Q(\cdot)$ due to the discretization, we prove in our Theorem 2 that the discrete action sequence can bring the optimal structure of LoCaL that can be simplified to obtain the true equation. This procedure is illustrated in lines 189-195.
> > >
> > > In general, we extend the discrete actions to continuous space and utilize a discretization procedure to reobtain the discrete values.

---

> > > > ### Author Response · Authors · 2022-08-09
> > > > **Reply to Reviewer ipq6, 2nd round, part 2**
> > > >
> > > > $\textbf{Q3: A thorough re-evaluation to make core ideas more digestible.}$
> > > >
> > > > Answer: Thanks a lot for your suggestions. We highly appreciate the reviewer's advice and re-evaluate the presentation of the paper. Specifically, we make the following efforts to let the readers easily capture the ideas and the procedures. First, we summarize our core ideas as follows. $(i)$ We decompose the whole SR problem into searching the symbol connections and estimating the symbol coefficients. To link them for a complete SR solution, we propose LoCaL with unknown structures and weights. $(ii)$ To estimate the weights of LoCaL, we show the LoCaL has local convexity in the loss surface given correct structures and some mild assumptions. $(iii)$ To search the structure of LoCaL, we develop a double convex deep Q-learning with provable search results. Second, we illustrate how we modify the paper in a top-down structure to illustrate each of the core ideas.
> > > >
> > > > To illustrate the core idea $(i)$, we add lines marked in blue in Subsection 3.1. Specifically, in the first paragraph of Subsection 3.1, we introduce LoCaL as a link for the two sub-problems (lines 95-97). We further demonstrate that (1) LoCaL has the capacity to represent the true equation in the second paragraph of Subsection 3.1 (line 102), and (2) LoCaL is the link for two sub-problems: the structures and weights of LoCaL are optimization variables of SR problem (line 116).
> > > >
> > > > To illustrate the core idea $(ii)$, we mention that LoCaL is proven to have local convexity in the first paragraph of Subsection 3.1 (lines 99-101).
> > > >
> > > > To illustrate the core idea $(iii)$, we add blue lines in Subsection 3.2. Specifically,
> > > >
> > > > (1) in the first paragraph of Subsection 3.2, we define the search states and actions for LoCaL (lines 129-131).
> > > >
> > > > (2) Then,  we conclude at the end of the first paragraph that a trajectory of the search results can identify the structure of LoCaL (lines 147-148).
> > > >
> > > > (3) We further theoretically summarize the search process as a Controlled Markov Process (CMP) that has optimal Q functions in the second paragraph of Section 3.2. This paragraph is added in response to reviewer 1 to maintain the soundness of our model.
> > > >
> > > > (4) In the third paragraph, we show the goal is to convexify the search optimization. To achieve the goal, we propose double convexity for the negative reward function and negative Q-function (lines 154-157).
> > > >
> > > > (5) In the fourth paragraph, we show how to convexify and train $-R(\cdot)$.
> > > >
> > > > (6) In the fifth paragraph, we show the convex function should be broadly defined over the continuous action space. We define the continuous actions and show it's doable to utilize a convex function to approximate the reward over the continuous space (lines 171-180). This validation is to reply to your $\textbf{Q1}$.
> > > >
> > > >  (7) With defined continuous action space, we similarly define the convex -$Q(\cdot)$ in the continuous action space in the sixth paragraph (lines 181-182).
> > > >
> > > >  (8) We also present the concrete optimization process in the remaining part of Subsection 3.2.
> > > >
> > > > We hope these specific illustrations can help the readers easily capture our logic and core contributions. In addition to the logic, we provide an overview of the technical framework with a high-level pseudo-code in Algorithm 1, Appendix A.1. This is based on the suggestion of Reviewer 1 to let the readers easily understand our algorithms. Please refer to the revised paper for our modifications. Thanks a lot for your appreciation and consideration.

---

### Official Review · Reviewer_AUni · 2022-07-19

**Rating:** 6
**Confidence:** 3
**Soundness:** 3 good
**Presentation:** 2 fair
**Contribution:** 3 good

**Summary:**

Authors propose to learn coefficients and symbols connections in the problem of Symbolic Regression (SR) by combining Q learning and neural networks. They rely on the convexity properties of of ICNN to ensure that the Q function and the rewards R() are convex wrt the state S and the action A of the MDP defined.

They use the convexity properties and the determinism of the MDP to derive several guarantees on their method - namely, that the exact symbols can be recovered for noiseless datasets because the problem is locally convex in a neighborhood of the true minimum.



**Questions:**

Could you clarify how W* is found ? By minimizing MSE with SGD over some randomly initialized W ?

l 202 : what "exact equations" mean ? In general there might be no unicity of the equations, so that true equation and the one returned by CONSOLE might be different.

Could you clarify what is the MDP exactly ? My whole understanding might be wrong ; but currently I feel like there is a recursive argument : a not well defined MDP is used to define a Q function (that should not exist) and a return (sum of the rewards received an the end of an episode - not markovian becasue states are too small and too poor to encode the equation being built), that is transformed into a convex reward R() given a ICNN surrogate, which is itself (using Theorem 1) used to explain that the aforementioned MDP is well defined (?) and that its Q function is convex.

Proof of theorem 4 (in appendix): authors uses the trick "x = s exp(log (|x| ))" with s the sign of x. This is always dangerous : s is not a constant, it it depends of x so one should write s(x) instead. Authors proceed with the proof for a fixed (chosen in advance) x_i until line 578 : then they introduce the objetcs y' and y'' which are defined for x_1..x_N. Since the notation s(x) is not used, it is not obvious (by reading the proof) that the differents values of s have been taken into account. I am not saying it is false, but it is worth explaining in the proof whether or not this is correct. This trick is sometimes fishy because of that.



**Limitations:**

With my current understanding of the paper, the whole theoretical framework used in the paper is invalid, which seriously impedes its impact. Moreover the experiments seem to be very small, specially when there is only linear {x} symbol pools.

**Strengths And Weaknesses:**

Strengths:

- i) solving complex combinatorial problem with RL is always an interesting approach
- ii) using ICNN to enforce good properties (and particular that the minimum is uniquely defined and easy to find) is a good take-home message from the paper
- iii) the experimental part contains numerous tasks and comparisons against other SOTA approaches, with figures and examples
- iv) experimental results are convinving about the eficiency of the method - even though I am not familiar with the field

Weaknesses:

major issues:

The whole approach consists in using Reinforcement Learning (RL), and more specifically Q-learning (and its neural network variants). Howerver RL requires to work on a Markov Decision Process (MDP), the Markov property must hold : P(s_2 | s_1, a_1) = P( s_2 | s_1, a_1, s_0, a_0). However it seems that the object defined here is not a proper MDP. Indeed, since the state s belongs to {0, 1} ^n_k we only have access to the activations of the layer k : no information about the previous activations, not mentioning the input x itself. Here, the state s has no knowledge of how "deep" we are in the equation being built. It feels weird : how a neural network can take the right actions if it does not "see" the whole input ? Moreover, in MDP, the reward depends of the current state s and action a, or of the current state s and future state s' (depending of the literature). This is not fulfilled by the construction given by the authors in either case.

If the problem is not a MDP, the Q function has no reason to exist, and fixed point based algorithms like value iteration (and variants such as Deep Q learning) are not guaranteed to converge. Even if the optimal Q function exists, it has no reason to be convex in general. This is not a property to be enforced with ICNN, it is a property of the MDP itself, that must be defined as such (Markov property must be fufilled !).

As said by authors in literature review, the problem of SR is NP hard. Hence, an attempt to convexify the problem that would yield polynomial time algorithm would fail. So either the problem cannot be cheaply convexified, either it induces a bias that prevents to find the global optimum of the original problem.

Even if it was a MDP, there are other issues. For example, the complexity of value iteration is O(S² * A) with A and S the action and space states respectively. For S defined by the authors, there is only 2^n_s states and 2^(n_s * n_s) actions. The total complexity of value iteration would be O( 2^(n_s  * n_s)). How does it compare to Deep Q learning in sample efficiency ? For dataset (1) the pool of activations is small, for (2) and (3) the pool of activations is only a singleton (!) the number of variables in input is also very small for (1) : only 3, while for (2) it is 5 nodes and for (3) it is 10 nodes. On those toy examples, the search space is so small that value iteration seems sufficient, no need for Deep Q learning

Theorem 1 is false. Since it is not a MDP, Q function is not defined. There is no reward in this MDP, only a convex surrogate function R() paramatrized by ICNN, that looks like the reward R received at each time step. But in reality, this R is not a Markvovian reward (that depends of state/action pair), it is a surrogate that depends of the final return of the trajectory.

Forcing the surrogate R and the network Q to be convex using ICNN induces a bias, but can not be used to prove anything about the underlying MDP.

l192 : "Then, we 192 prove that the convexity of the negative optimal Q-function eventually yields the exact equations."
Well, then, why is there a non zero error in experiments Fig. 3 ? Such error is not negligible.

minor issues:

i) It it not clear what neural networks bring here. In general their power in combinatorics is their generalization capabilities - i.e their capacity to interpolate the value of unseen inputs. Paper lack details about this precise contribution of NN.

ii) Theorem 4 is very ad-hoc due to its complicated form. i,j and k variables are not defined. Even the appendix does not contain a clear statement o the theroem, only the proof. Hence, it is hard to understand what the objets clearly are.

l 202 : what "exact equations" mean ? In general there might be no unicity of the equations, so that true equation and the one returned by the network might be different.

iii) theorem 2 is assumes that the optimal W* is recovered (why ? How ?) and that the sequence of actions is selected according to the optimal policy Q*, assuming it exists, such that the optimum of Q* corresponds to exact equations (using the fact that the dataset is noiseless). Authors conclude the exacts equations can be recovered. This is essentially trivial : authors are concluding that under the hypothesis that the optimum exists and has been found, it has beend found...

iv) theorem 3: it seems that authors are saying that in the neighborhood of the minimum, if for one of the examples there is no plateau (assumption 2), the second orer derivative (assuming it exists, assumption 1) is positive. For unconstrained minimization of twice differentiable function, this is a well known property: https://math.stackexchange.com/a/19473 (second paragraph)

---

> ### Author Response · Authors · 2022-08-02
> **Response to Reviewer AUni, model soundness**
>
> We are really grateful for the reviewer's insightful comments. The reviewer's central concern for these questions can be summarized as follows: ``Because (1) the state transition in our setting is non-Markov, and (2) the reward in our setting is non-Markov, the problem setting we studied is not a valid MDP, and thus our subsequent theoretic results collapse". For (1), we would like to point out that this is a factual misunderstanding, i.e., our state transition indeed satisfies Markov property. For (2), the reviewer is right in that our reward is non-Markov. Although some existing works in Neural Architecture Search (NAS) [1] and Symbolic Regression [2] still treat this as an MDP, as discussed in the paper, your question does inspire us to further check the validity of our theoretic results. After careful and thorough investigation, we are reassured that all of our theoretic results hold firmly with the non-Markov reward. The main reason is that our search process can be defined as a Controlled Markov Process (CMP), which is a Markov Decision Process (MDP) without a reward function [3]. Then, given a CMP with rewards that can evaluate the state-action trajectories, the optimal Q function still exists. We provide the proof sketch as follows and have revised the paper accordingly (the full proof will be added to the final version of the paper). We invite the reviewer to check these proof sketches and we look forward to having further discussions with the reviewer.
>
> $\textbf{Markov state transition.}$ First, we show our state transition satisfies the Markov property. Specifically, Equation (1) in our paper shows that the next state $\boldsymbol{s}\_{k+1}$ equals the matrix multiplication between the current state $\boldsymbol{s}\_k$ and the matrix $\boldsymbol{Z}\_k$ that is a matricization of the current action $\boldsymbol{a}\_k$, where $k$ is the index of the state. Therefore, the state transition satisfies the Markov property. Similar settings exist in Neural Architecture Search (NAS) [1] and Symbolic Regression [2] as discussed in our paper.
>
> $\textbf{Controlled Markov Process}$. Due to the Markov property of the state transition, we define our search process as the Controlled Markov Process (CMP) [3,4]. By the CMP definition [4], our CMP is composed of our state, action, state transition probability, a discounter factor $\gamma$, and a start state (i.e., $\boldsymbol{s}\_0$ in Equation (1)). In general, CMP is a Markov Decision Process (MDP) without a reward function [3].
>
> $\textbf{Trajectory Ordering (TO) and our reward $R(\cdot)$}$. For one CMP, TO ranks trajectories of state action pairs [3]. In our paper, we define the trajectory from $(\boldsymbol{s}_0,\boldsymbol{a}_0)$ to $(\boldsymbol{s}\_{K-1},\boldsymbol{a}\_{K-1})$ for $K$-layer Locally Convex Equation Learner (LoCaL). Then, our reward function $R(\cdot)$ realizes a TO for our defined trajectories [3] since the ordering of trajectories can be determined by $R(\cdot)$. More specifically, $R(\cdot)$ is trained with the end-of-trajectory reward $R_t$ for the $t^{th}$ trajectory in our paper (lines 146-153 in the first version of the paper) and can rank trajectories.
>
> $\textbf{Existence of a reward bundle.}$. A reward bundle is an automation-like structure to produce rewards for a CMP [4]. By Corollary 2 of [4], there exists a reward bundle for our defined CMP and TO realized by $R(\cdot)$.
>
> $\textbf{Existence of the optimal Q-function.}$ We pair our CMP with the reward bundle to form a Split Partially Observable MDP (Split-POMDP) [4]. Then, by Proposition 1 and Corollary 1 in [4], our Split-POMDP will always have an optimal deterministic policy that *only* depends on states in our CMP. By the proof of Proposition 1 in [4], the optimal policy optimizes the value function over states in CMP. Further, the value function is an evaluation of trajectories for our TO by the proof in Corollary 2 in [4]. Additionally, our TO is realized by our proposed reward function $R(\cdot)$. Therefore, the optimal Q-function exists for our CMP and our proposed $R(\cdot)$. The existence of the optimal Q-function lays the foundation for our Theorem 1 in the paper.
>
> We have modified the statement of Theorem 1 and claim the existence of optimal Q-function. In the meantime, we have added the above proof sketch to the proof of Theorem 1. The detailed proofs will be added to the final version of the paper. We invite the reviewer to check these proof sketches and we welcome the reviewer to have further discussion with us.
>
> [1] Baker, Bowen, et al. "Designing neural network architectures using reinforcement learning." 2016.
>
> [2] Petersen, Brenden K., et al. "Deep symbolic regression: Recovering mathematical expressions from data via risk-seeking policy gradients." 2019.
>
> [3] Abel, David, et al. "On the expressivity of Markov reward." 2021.
>
> [4] Abel, David, et al. "Expressing Non-Markov Reward to a Markov Agent." 2021.

---

> > ### Author Response · Authors · 2022-08-02
> > **Response to Reviewer AUni, other questions, part 1**
> >
> > We summarize and answer the remaining questions from the reviewer.
> >
> > $\textbf{Q1: State space and the ability to consider past information.}$
> >
> > Answer: We want to clarify that the state $\boldsymbol{s}$ does *not* belong to $\\{0, 1\\}^{n\_s}$. For example, in Fig. 1 in the paper, we can calculate $\boldsymbol{s}\_2=[2,1,1,0,0,0]^T$ by Equation (1). Second, Equation (1) in the paper implies that the current state $\boldsymbol{s}\_k$ is the result of a series of multiplications of initial state $\boldsymbol{s}\_0$ and past actions $\\{\boldsymbol{a}\_0,\boldsymbol{a}\_1,\cdots,\boldsymbol{a}\_{k-1}\\}$. Thus, all the information in the previous layers is considered to enable the decision-making for the current action.
> >
> > $\textbf{Q2: Bias in the problem.}$
> >
> > Answer: We agree that we could introduce biases in the convexification process. For example, we have assumptions (1)-(4) to prove the local convexity of our LoCaL model. However, these assumptions are admissible for many underlying equations as explained in the last paragraph in Section 4.
> >
> > $\textbf{Q3: Deep Q learning sample efficiency vs. value iteration complexity.}$
> >
> > Answer: There are many studies to improve the sample efficiency in deep Q-learning by designing novel methods to manage the trade-off between exploration and exploitation [5-8]. In our case, we utilize $\epsilon$-greedy method and the regret complexity (a measure of the RL performance of using samples) is $\Omega(\min\\{KT,2^\{Kn_s^2/2}\\})$, where $n\_s$ is the dimension of the state space, $T$ is the number of episodes, $K$ is the number of steps in each episode (i.e., number of layers in our LoCaL) [5]. Therefore, this complexity is comparable to the value iteration complexity $O(2^{n_s^2})$.
> >
> > $\textbf{Q4: The need for Deep Q learning in experiments.}$
> >
> > Answer: Since $n_s=\max\\{n_k\\}_{k=0}^K$ in our paper, we can list the experimental values of $n_s$ as follows: for two synthetic datasets in (1), $n_s=9(15)$ as the number of neurons in the activation layer in LoCaL, respectively. For dataset (2), $n_s=50$ as the number of neurons in the multiplication layer in LoCaL. For dataset (3), $n_s=10$ as the number of neurons in the activation layer in LoCaL. The number of possible actions ranges from $2^{9\times9}=2^{81}$ to $2^{50\times50} = 2^{2500}$, which can hardly be tackled using tabular Q learning. Thus, deep Q learning is a possible solution. In addition, deep learning-based RL is also well-motivated in other work in Symbolic Regression [2,9], as discussed in the paper.
> >
> > $\textbf{Q5: Non-negligible errors in experiments.}$
> >
> > Answer: Our proof only shows that the negative optimal Q-function can bring the exact equations. However, given fixed episodes $T=600$ for fair comparisons, the learned Q function may not converge to the optimal Q function, which causes small but non-negligible errors. However, we emphasize that our method is still much better than state-of-the-art symbolic regression methods.
> >
> > $\textbf{Q6: Roles of Neural Networks (NNs).}$
> >
> > Answer: In our paper, NN has two roles. First, in the discrete optimization to search for correct symbols and symbol connections to formulate the equation, we utilize NN (i.e., Input Convex Neural Network (ICNN) [10]) to convexify $-Q(\cdot)$ and $-R(\cdot)$ such that they both take in continuous actions, which is essentially an interpolation. Then, the convex optimization can happen over the continuous action space, which leads to provable results with mild assumptions. Second, with the search result, we build another NN, i.e., Locally-Convex Equation Learner (LoCaL). We prove that under some mild assumptions, the loss surface of LoCaL has strictly and locally convex regions within the global optimal solution. Then, the training of LoCaL can lead to true coefficients of symbols (i.e., parameters of LoCaL) in the symbolic equations.
> >
> > $\textbf{Q7: No theorem statement in the proof.}$
> >
> > Answer: Thanks a lot for your suggestion and we have re-stated the theorems in the appendix to let the readers easily understand the object. Please check the revised revision.
> >
> > [2] Petersen, Brenden K., et al. "Deep symbolic regression: Recovering mathematical expressions from data via risk-seeking policy gradients." 2019.
> >
> > [5] Jin, Chi, et al. "Is Q-learning provably efficient?." 2018.
> >
> > [6] Gu, Shixiang, et al. "Continuous deep q-learning with model-based acceleration." 2016.
> >
> > [7] Yavas, Ugur, et al. "A new approach for tactical decision making in lane changing: Sample efficient deep Q learning with a safety feedback reward." 2020.
> >
> > [8] Li, Gen, et al. "Breaking the sample complexity barrier to regret-optimal model-free reinforcement learning." 2021.
> >
> > [9] Landajuela, Mikel, et al. "Improving exploration in policy gradient search: Application to symbolic optimization." 2021.
> >
> > [10] Amos, Brandon, Lei Xu, and J. Zico Kolter. "Input convex neural networks." 2017.

---

> > > ### Author Response · Authors · 2022-08-02
> > > **Response to Reviewer AUni, other questions, part 2**
> > >
> > > $\textbf{Q8: ''exact questions" in Theorem 1.}$
> > >
> > > Answer: We agree that ``exact equations" may cause confusion and thanks a lot for pointing this out. Our intention is to state that the learned equation in the network can be simplified to the true equation $g(\cdot)$ defined in the paper. Thus, we have changed the statement ''$f^{\*}(\cdot;W^{\*})$ represents the exact equations." to ''$f^{\*}(\cdot;W^{\*})$ can be simplified to the true equation $g(\cdot)$".
> > >
> > > $\textbf{Q9: Why and how $W^{\*}$ can be recovered in Theorem 2.}$
> > >
> > > Answer: The assumption that $W^{\*}$ is recovered can be verified by our Theorems 3 and 4. Specifically, Theorem 3 concludes that for the correct search result, i.e., $(\boldsymbol{s}\_0,\boldsymbol{s}\_1^{\*},\cdots,\boldsymbol{s}\_K^{\*})$ and $(\boldsymbol{a}\^{\*}_0,\boldsymbol{a}\_1^{\*},\cdots,\boldsymbol{a}\_{K-1}^{\*})$ obtained via the optimal Q function, there is a locally and strictly convex region over the loss surface of LoCaL. Theorem 4 further quantifies the region. By our analysis in the last paragraph of Section 4, the size of the region is large for many scientific and engineering equations. Therefore, a proper initialization can lead to the training of LoCaL to find the global optimal parameters ($W^{*}$) using gradient descent.
> > >
> > >  We minimize the MSE of LoCaL by the Adam algorithm to obtain the weight of LoCaL $W^{*}$. Random initialization of $W$ is OK as long as the initial guess lies within the locally convex region that has a relatively large range. In this paper, we do not discuss the scenario when the initial guess is beyond the region. However, it's an interesting direction to develop an effective strategy to encourage the initial guess to lie in the proper regions.
> > >
> > > $\textbf{Q10: Theorem 2 is trivial.}$
> > >
> > > Answer: Theorem 2 is not trivial since we cannot directly conclude that the optimal state/action sequences from $Q\^{\*}(\cdot)$ can lead to the exact equation. More specifically, $Q\^{\*}(\cdot)$ and $Q(\cdot)$ directly depend on our designed reward function $R(\cdot)$ but not the end-of-trajectory reward. Thus, we can not directly say the optimum of $Q\^{\*}(\cdot)$ corresponds to the true equations and need to provide proofs.
> > >
> > > $\textbf{Q11: Theorem 3 is well-known.}$
> > >
> > > Answer: This is a factual misunderstanding and we would like to provide the following clarifications. Our assumptions (1) and (2) in Theorem 3 restrict the properties of the first/second-order derivatives of the true equation $g(\boldsymbol{x})$. This is not the property of the objective function in the unconstrained minimization. More specifically, our objective of training LoCaL is $L(W)=\frac{1}{2N}\sum_{i=1}^N(\boldsymbol{y}_i-f(\boldsymbol{x_i};W))^2$, where $N$ is the number of training samples, $\\{\boldsymbol{x}_i,\boldsymbol{y}_i\\}\_{i=1}^N$ is the training dataset, and $f(\cdot;W)$ is the map function of LoCaL. Thus, we cannot directly conclude that $L(W)$ is locally convex in $W$.
> > >
> > > $\textbf{Q12: Dangerous use of the trick in the proof of Theorem 4.}$
> > >
> > > Answer: This is a factual misunderstanding and we would like to provide the following clarifications. First, our defined $\boldsymbol{S}$ matrix is not a sign matrix but a selection matrix to select the neurons used as multiplicative terms in the multiplication layer in LoCaL, as explained in lines 560-561 in the first version of the paper. Thus, $\boldsymbol{S}$ is a constant matrix and is essentially the search result of our RL agent. Therefore, our following proofs with respect to $\boldsymbol{S}$ will not suffer the risk you mentioned. Second, we define our sign vector $\boldsymbol{s}$ in lines 565 to 567. However, as claimed in lines 563-568, we utilize the equation $\hat{y}(\boldsymbol{x},W)=\boldsymbol{W}_1^T\exp(\boldsymbol{S}^T\log(\Phi(\boldsymbol{W}_0^T\boldsymbol{x})))$ to take place of the equation $\hat{y}(\boldsymbol{x},W)=\boldsymbol{W}_1^T\boldsymbol{s}\circ\exp(\boldsymbol{S}^T\log(|\Phi(\boldsymbol{W}_0^T\boldsymbol{x}|)))$ in the following derivations, where the notations are defined in the paper. This is doable since they both produce the same values of gradients with respect to $\boldsymbol{W}_0$ and $\boldsymbol{W}_1$, and the following derivations only require these values. Therefore, the sign vector $\boldsymbol{s}$ is removed in the following derivation.

---

> > > > ### Comment · Reviewer_AUni · 2022-08-05
> > > > **Thank you for your clarifications**
> > > >
> > > > I'd like to thank the authors for their reactivity and their precise answers to my questions. My initial review that was lengthy and not well organized, I acknowledge that must have been particularly challenging for the authors to address my concerns in a concise manner.
> > > >
> > > > My initial "strong reject" rating was motivated by the fact the paper was not sound. I appreciate that the authors found a way to make it more sound. However, as I do not master the tools used to correct the issue, I cannot confidently say the solution is correct. As a consequence I increase my rating and I lower my confidence.
> > > >
> > > > I still have issue with the MDP however; that I will detail below. I incline to increase my rating again upon satisfactory answer of the authors regarding this concern.
> > > >
> > > > #### **Q1 (state space and ability to consider past information)**
> > > > My concerns regarding the states are detailed in my next message. On this particular issue I also would appreciate to read the opinion of other reviewers.
> > > >
> > > > #### **Q2**
> > > > If my understanding is correct, the last paragraph of Section 4 is about the (local) convexity of the problem of learning $W$ coefficients. This is a separate problem from the convexification of $R(s,a)$ and $Q(s,a)$ wrt $s,a$
> > > >
> > > > #### **Q3,Q4,Q6**
> > > > Thanks to those arguments I am now convinced that Tabular Q learning or Value Iteration wouldn't have been tractable.
> > > >
> > > > #### **Q5**
> > > > Thank you for the precision.
> > > >
> > > > #### **Q7**
> > > > Thank you; the reading is more pleasant this way.
> > > >
> > > > #### **Q8**
> > > > Thank you for this modification.
> > > >
> > > > #### **Q9**
> > > > Thank you for the precisions. I believe the paper would gain clarity if you could write this precisely somewhere: lines 213-214 are a good start, but IMO this part of the paper would benefit from its own paragraph. Or maybe even in lines 153-160 where you introduce the " end-of-trajectory reward" computed from the "the trained $f_t(\cdot)$ function ". It is not clear for me that the coefficients W are fit with MSE at the end of episode.
> > > >
> > > > #### **Q10, Q11**
> > > > Sorry for having stated the theorem was trivial. In light of your precisions I understand what they bring to the paper.
> > > >
> > > > #### **Q12**
> > > > Thank your for your precisions regarding $s$.

---

> > > > > ### Comment · Reviewer_AUni · 2022-08-05
> > > > > **Soundness of the MDP**
> > > > >
> > > > > As mentionned before, I would appreciate insights from other reviewers to ensure that my understanding of the issue is correct.
> > > > >
> > > > > ## Summary of the method
> > > > >
> > > > > Because the code is not given, shady parts of the algorithm are not easy to infer solely from the content of the paper itself.
> > > > >
> > > > > I feel the need to clarify here my understanding of the method, to ensure we are talking about the same thing. I wrote below a sequence of statements regarding your algorithm. Could you specify which ones are false (if any) ?
> > > > >
> > > > > ### LoCaL structure
> > > > >
> > > > > a) LoCaL is a sequence of structures like the one depicted Figure 1.
> > > > >
> > > > > b) In Figure 1 there is only one block, but in general LoCaL can contain an arbitrary high number of such blocks.
> > > > >
> > > > > c) The first layer of LoCaL is the set of variables of the problem: $h_0=x$
> > > > >
> > > > > d) Each block contains 3 layers: activation layer, multiplication layer, addition layer
> > > > >
> > > > > e) Each layer $k$ is connected to the following layer with a matrix binary matrix $Z_k$ that encodes active connections
> > > > >
> > > > > f) For symbolic activation and summation laye we have $h_{k+1}=Z_k^T\circ W_k^Th_k$ : the output of each layer $k$ is multplied by weights $W_k$ before being selected with $Z_k$. The parameters are $W_k$.
> > > > >
> > > > > g) For symbolic multiplication there is no parameters $W_k$ involved, only $Z_k$ matrix to select terms in multiplication.
> > > > >
> > > > > ### MDP learning
> > > > >
> > > > > The MDP is defined as follow:
> > > > >
> > > > > a) The combinatorial problem associated to the choice of $Z_k$ is turned into RL problem by mapping $Z_k$ to an action $a_k$ : a discrete action space
> > > > >
> > > > > b) A trajectory is a sequence of $K$ actions $a_k$, where $K$ denotes the depth of LoCaL.
> > > > >
> > > > > c) $s_0$ is vectors $[1, 1, ..., 0]$ where the number of $1$ correspond to the number of variables, and the zeros $0$ are here for padding.
> > > > >
> > > > > d) Each state $s_k\in\mathbb{Z}^{n_s}$ that that $(s_k)_i$ contains the number of incoming edges of neuron $i$ in layer $s_k$. There is also zero padding to account for discrepancy in the number of neurons of each layer.
> > > > >
> > > > > ### Final algorithm
> > > > >
> > > > > a) The agent plays a sequence of actions in the MDP (i.e select some connections $Z_k$ with actions $a_k$). This trajectory yields a symbolic equation with parameters $W_k$. Technically, this should be considered as a CMP (not an MDP) because there is not reward yet.
> > > > >
> > > > > b) At the end of a single trajectory, the coefficients $W_k$ are fitted by minimizing MSE with Adam on the train set $\{x_i,y_i\}$.
> > > > >
> > > > > c) The NRMSE obtained after fitting coefficients is turned into a "end of trajectory reward", that allows you to rank trajectories (TO).
> > > > >
> > > > > d) The paper you cite [32] explains that TO does not necessarily yields markovian rewards. But by results of paper [31] (Corollary 2) there exists a reward bundle that captures this ordering. Note that a reward bundle is not necessarily a markovian reward (wrt the states of the CMP, because it is markovian wrt to the states of the reward bundle by definition).
> > > > >
> > > > > e) Because there is a reward bundle, you obtain a Split-MDP. The agent does not have access to the reward state of the bundle, so it is in fact a Split-POMDP.
> > > > >
> > > > > f) In paper [31], Prop 1. concludes that a reward bundle exists for a set of deterministic policies $\Pi_g$ chosen in advance, but you don't specify which set you consider here. Similarly, Corollary 1 of [31] assumes that $\Pi_g$ is given. This is crucial because the optimal policy that depends only of CMP states is built upon the bundle defined in proof of Prop 1, which itself depends of the set $\Pi_g$ chosen. I couldn't find the appendix of the paper [31] you cite, so I cannot attest that the proof of Corollary 2 in [31] allows to conclude that your convex reward R() allows trajectory ordering and Q-learning.
> > > > >
> > > > > g) Putting appart technical difficulties of (f), you chose to parametrize $R(s,a)$ with an ICNN (convex wrt to $s,a$) by using the RNMSE of your fitted $f_t()$.
> > > > >
> > > > >  h) You learn the Q function associated to this non Markovian reward (because it is still TO) with Q learning. Your Q function is still parametrized as an ICNN.
> > > > >
> > > > > i) You use Deep Q learning to learn Q, with epsilon greedy exploration.
> > > > >
> > > > > j) You conclude that the optimal policy $a_k=\text{argmax}_a Q(s_k,a)$ yields the "correct equations", and $W_k$ is learned easily with Adam from those correct equations.
> > > > >
> > > > > ## Remaining questions
> > > > >
> > > > > > Second, Equation (1) in the paper implies that the current state $s_k$ is the result of a series of multiplications of initial state $s_0$ and past actions $\{a_0,a_1,...,s_{k-1}\}$.
> > > > >
> > > > > It is not because it is the result of serie of multiplications that $s_{k-1}$ necessarily contain information about previous states. In particular if $Z_k$ is not invertible.
> > > > >
> > > > > For example, with a different conection pattern in Fig 1. we could have $s_2=[1, 1, 0, 0, 0, 0]$ for an equation that do not use multiplications neither terms involving $x_2$. How to distinguish it from $s_0$ in your replay buffer to learn $Q$ ?

---

> > > > > > ### Author Response · Authors · 2022-08-06
> > > > > > **The second round response to Reviewer AUni, part 1**
> > > > > >
> > > > > > Thanks a lot for the reviewer's timely feedback and insightful questions. The reviewer still has concerns on two key questions: 1) the construction of the reward bundle in the proof and 2) and the case of repeated states in our CMP. Thus, we show how the reward bundle can be constructed over the trajectory space and how our algorithm can handle the repeated state scenario. We also address other comments and slightly modify our paper. Please feel free to check our modified version.
> > > > > >
> > > > > > $\textbf{Q1: The convexification of $-R(\boldsymbol{s},\boldsymbol{a})$ and $-Q(\boldsymbol{s},\boldsymbol{a})$ may cause bias.}$
> > > > > >
> > > > > > Answer: We agree that the convexification for the $-R(\boldsymbol{s},\boldsymbol{a})$ and $-Q(\boldsymbol{s},\boldsymbol{a})$ also induces bias. Specifically, the convexity of $-R(\boldsymbol{s},\boldsymbol{a})$ and $-Q(\boldsymbol{s},\boldsymbol{a})$ is an additional constraint that never appears in the vanilla deep Q-learning. Such a constraint may hurt the convergence to the optimal Q-function. However, we prove in Theorem 1 that our deterministic and linear state transition process can guarantee that the optimal Q-function is also convex. Thus, the convexity constraint of $-Q(\boldsymbol{s},\boldsymbol{a})$ will not hurt the convergence.
> > > > > >
> > > > > > $\textbf{Q2: Please clarify the training process of LoCaL in the paper.}$
> > > > > >
> > > > > > Answer:  Thanks a lot for your suggestions. We follow your suggestions and add the description in both lines 154-160 and lines 213-214, respectively. For lines 154-160, we add the following statement. "Basically, we utilize the gradient method (e.g., Adam [1]) to train $f\_t(\cdot;W\_t)$ and obtain the optimal set of weights $W\_t^{\*}$ for the $t^{th}$ episode of LoCaL by minimizing the Mean Square Error (MSE). Then, we can calculate the Normalized Root-Mean-Square Error (NRMSE) [2] of the trained $f\_t(\cdot;W^{\*})$ such that $\text{NRMSE}\_t=\frac{1}{\sigma\_y}\sqrt{\frac{1}{N}\sum_{i=1}^N(\boldsymbol{y}\_i-f_t\big(\boldsymbol{x}_i;W\_t\^{\*})\big)^2}$, where $\sigma_y$ is the standard deviation of the outputs."
> > > > > >
> > > > > > For lines 213-214, we add the following statement "For the optimal search structure of LoCaL, i.e., $f\^\*(\cdot;W)$, the optimal weight set $W\^{\*}$ is also trained via gradient method like Adam [1]."
> > > > > >
> > > > > > [1] Kingma, Diederik P., and Jimmy Ba. "Adam: A method for stochastic optimization." arXiv preprint arXiv:1412.6980 (2014).
> > > > > >
> > > > > > [2] Petersen, Brenden K., et al. "Deep symbolic regression: Recovering mathematical expressions from data via risk-seeking policy gradients." arXiv preprint arXiv:1912.04871 (2019).

---

> > > > > > > ### Author Response · Authors · 2022-08-06
> > > > > > > **The second round response to Reviewer AUni, part 2**
> > > > > > >
> > > > > > > $\textbf{Q3: Clarification of the summary of the method.}$
> > > > > > >
> > > > > > > Answer: We are highly grateful for the reviewer's careful reading and summary. The summary is generally correct except for two errors. The reviewer also poses an important question in the section of Final Algorithm f). We answer the question and look forward to having further discussions with the reviewer.
> > > > > > >
> > > > > > > $\textbf{Corrections}$.
> > > > > > >
> > > > > > > In b) of the Section of LoCaL structure: we have two blocks in Fig. 1.
> > > > > > >
> > > > > > > In d) of the Section of MDP learning: the state value $(\boldsymbol{s}_k)_i$ for neuron $i$ cannot be defined as the number of incoming edges to the neuron $i$. For example, in Fig. 1, we can compute $\boldsymbol{s}_3=[2,2,0,0,0,0]^T$. While the first neuron in the summation layer only takes in $1$ edge, we have $(\boldsymbol{s}_3)_1=2$. Specifically, the previous layer (i.e., the multiplication layer) has the state $\boldsymbol{s}_2=[2,1,1,0,0,0]^T$. Thus, the first neuron in the multiplication layer has a larger state value (i.e., 2) than those of other neurons since this neuron has two incoming edges. To summarize, the state value $(\boldsymbol{s}_k)_i$ for neuron $i$ is a result of both the incoming edges and the previous edges related to neuron $i$.
> > > > > > >
> > > > > > > $\textbf{Questions in f) in the Section of Final algorithm}$.
> > > > > > >
> > > > > > > Answer: In Proposition 1 and Corollary 1 of [3], we agree it's crucial to define $\Pi_G$ that determines the $\textbf{reward state space}$ and the $\textbf{reward start state}$ in the reward bundle. For the treatment of Trajectory Ordering (TO) in Corollary 2, the proof shows that the $\textbf{reward state space}$ can be defined as the trajectory space, which is known in our case. Next, we can define the $\textbf{reward start state}$ as an initial set of sample trajectories. Then, the reward bundle can be successfully constructed for the case of TO, which is similar to the proof in Proposition 1 in [3]. Note that in the search process, we can $\textbf{repopulate}$ the sample trajectory set based on the new search result, as shown in the proof of Proposition 1 in  [3]. This re-population helps to find trajectories that are consistent with the state-action taken in one CMP [3]. Such a process can be repeated to prove the $\textbf{existence}$ of the optimal policy that determines the best trajectory in the trajectory space. We believe the above explanation is reasonable although we cannot find the full proofs online.
> > > > > > >
> > > > > > > We have already written an email to the authors of [3] in DeepMind to ask for the full proofs. Since [3] is public and is the supplementary material of the NeurIPS paper [4], we believe their theories are correct. Finally, based on the above explanations and the correctness of Corollary 2 in [3], the reward bundle exists in our TO realized by $R(\cdot)$.
> > > > > > >
> > > > > > > $\textbf{Q4: The scenario of repeated states.}$
> > > > > > >
> > > > > > > Answer: We really appreciate the reviewer's insightful question. In the setting proposed by the reviewer, our state definition cannot distinguish from $\boldsymbol{s}_0$ to $\boldsymbol{s}_2$ as they have the same values. However, our proposed algorithm still has the capacity to produce the correct search results even if $\boldsymbol{s}_0$ and $\boldsymbol{s}_2$ have different optimal actions. Specifically, when $\boldsymbol{s}_0=\boldsymbol{s}_2$, we can still have $\textbf{different global optimal solutions}$ of $\arg\max\_{\boldsymbol{a}} Q(\boldsymbol{s}\_0,\boldsymbol{a})$ given an ICNN [5] to approximate $-Q(\cdot)$ with high representational power. Further, in our Algorithm 1, we solve this optimization via the so-called bundled entropy method [5] (lines 174-175 in the first round of the revised paper) which requires different initializations (e.g., we conduct $10$ initial guesses to solve the convex optimization in our experiments). This shows that with enough search time, our Algorithm 1 has the capacity to find different optimal actions $\boldsymbol{a}\_0\^{\*}$ and $\boldsymbol{a}\_2\^{\*}$ for $\boldsymbol{s}\_0$ and $\boldsymbol{s}\_2$, respectively, which can build the LoCal that corresponds to the true equation. Additionally, based on the terminate criterion in our Algorithm 1, the search process will only terminate when $R_t\approx 1$, which means the true equation is found.
> > > > > > >
> > > > > > > Another way is to directly remove the repeated-state scenario by appending the state index (i.e., the layer index in LoCaL) as an additional entry to the state vector. The linear transformation in Equation (1) in the paper still holds as long as we add a constant vector $[0,\cdots,0,1]^T$ for the index entry. We treat this modification as our model variation. We will release both the original and the variant models to Github after the well-organization of our codes.
> > > > > > >
> > > > > > > [3] Abel, David, et al. "Expressing Non-Markov Reward to a Markov Agent." 2021.
> > > > > > >
> > > > > > > [4] Abel, David, et al. "On the expressivity of markov reward." 2021.
> > > > > > >
> > > > > > > [5] Amos, Brandon, et al. "Input convex neural networks."  2017.

---

> > > > > > > > ### Comment · Reviewer_AUni · 2022-08-06
> > > > > > > > **Concluding discussion**
> > > > > > > >
> > > > > > > > I'd like to thank the authors once again for this interesting discussion.
> > > > > > > >
> > > > > > > > ### Q1
> > > > > > > > I am now convinced, thank you.
> > > > > > > >
> > > > > > > > ### Q2
> > > > > > > > Thank you for this modification.
> > > > > > > >
> > > > > > > > ### Q3
> > > > > > > >
> > > > > > > > d) If my understanding is correct, then $(s_k)_i$ contains the number of *incoming paths* from the input to neuron $i$ in layer $k$. I believe it can be proved by induction over indices $k$. If it is true, it should be written clearly.
> > > > > > > >
> > > > > > > > ### Q4
> > > > > > > >
> > > > > > > > > we can still have **different global solutions** of $\text{argmax}_a Q(s_0,a)$
> > > > > > > >
> > > > > > > > This is surprising ; if $Q(s,a)$ is convex wrt $a$ then either the optimum is unique, either it is a plateau (i.e the set $\text{argmax}_a Q(s_0,a)$ is convex) and no local minimum exists.
> > > > > > > >
> > > > > > > > > which requires different initializations (e.g., we conduct initial guesses to solve the convex optimization in our experiments)
> > > > > > > >
> > > > > > > > For the same reason I am surprised that you rerun 10 times the *bundle entropy method* with different initializations: aren't you sure the algorithm does not return the same solution every time ?
> > > > > > > >
> > > > > > > > > we solve this optimization via the so-called bundled entropy method [5] (lines 174-175 in the first round of the revised paper)
> > > > > > > >
> > > > > > > > We shall not confuse the solution of $\text{argmax}_a Q(s_0,a)$ and the one returned by an algorithm that aims to solve this problem.
> > > > > > > >
> > > > > > > > To conclude, playing exactly $a_k=\text{argmax}_a Q(s,a)$ does not allow you to chose different actions for $s_0$ and $s_2$ if $s_0=s_2$.
> > > > > > > >
> > > > > > > > But what happens is that you are **not** playing $a_k=\text{argmax}_a Q(s,a)$ .
> > > > > > > >
> > > > > > > > You are playing $a_k=f_{\epsilon}(Q,s_k)$ where $f_{\epsilon}$ *attempts* to solve $\text{argmax}_a Q(s,a)$ with *bundle entropy method*, and sometimes pick an action $a_k$ at random with probability $\epsilon$.
> > > > > > > >
> > > > > > > > The intrinsic stochasticity of $f_{\epsilon}$ is what allows you to find the optimal trajectory with non zero probability. So the role of $\epsilon$, in your case, extends further than just exploration.
> > > > > > > >
> > > > > > > > For example, assume that an oracle gives you the optimum $Q*$. Then if you play deterministically $a_k=\text{argmax}_a Q*(s,a)$ you will get $a_0=a_2$ whenever $s_0=s_2$. This proves that $\epsilon$ is not only exploration, but also a tool that ensures diversity of trajectories.
> > > > > > > >
> > > > > > > > > Another way is to directly remove the repeated-state scenario by appending the state index (i.e., the layer index in LoCaL) as an additional entry to the state vector.
> > > > > > > >
> > > > > > > > Thank you, this is what I wanted to see from the begining. __I currently believe this is the only way to make your MDP more sound and this is the reason I am inclined to increase my rating to "borderline accept" or more.__
> > > > > > > >
> > > > > > > > Also, a enhenced state $\tilde s=[s,\text{index}]$ is stricly more expressive than a state $s$, because it allows more complex optimal policies.
> > > > > > > >
> > > > > > > > I expect performance gains on your method if you include the index (or a normalized version between $[0,1]$, or a binary encoding, etc.. whatever ensures the training remains stable) in your state. Or maybe just faster convergence. Those additional experiments are crucial.
> > > > > > > >
> > > > > > > > ## Change of rating
> > > > > > > >
> > > > > > > > Please find below my final rating:
> > > > > > > >
> > > > > > > > * __I changed the *soundness* from "fair" to "good"__, since theoretical issues seem to have been addressed in satisfactory manner,
> > > > > > > > * __I changed the *contribution* from "fair" to "good"__, since the approach is original and now better theoretically grounded,
> > > > > > > > * __I keep the *presentation* at "fair".__ The paper has some merits regarding its presentation (figures, etc... ) on a topic that requires a lot of tools and background. However it took me some time to fully understand the ideas. I suggest the authors to look at my summary to improve the organization of the paper. I suggest adding a high level pseudo code for example. No need for precise details like the peudo code of Deep Q learning, but at least a high level overview of your strategy.
> > > > > > > > * Now that I am comforted in my understanding of the paper __I am increasing my confidence from 2 to 3__
> > > > > > > > * __I decided to increase my rating to "borderline reject"__ : the ideas of the paper merits some exposure but the current presentation harm them.
> > > > > > > >
> > > > > > > > Please find below actionable items to increase your rating:
> > > > > > > >
> > > > > > > > (a) the modifications you promised;
> > > > > > > > (b) a pseudo code;
> > > > > > > > (c) model variation with state index: my rating will be flipped to 5 (or more, depending on presentation) if this experiment is done.
> > > > > > > >
> > > > > > > > Ordinary I do not like asking for more experiments since it is the common reproach we find in Neurips reviews and burden the authors during rebuttal. However, in your particular case, the soundness depends on it.
> > > > > > > >
> > > > > > > > Thank you once again for this challenging debate and good luck regarding those experiments.

---

> > > > > > > > > ### Author Response · Authors · 2022-08-06
> > > > > > > > > **Thanks a lot for your comments**
> > > > > > > > >
> > > > > > > > > Thanks a lot for your comments. We agree that utilizing $\tilde{\boldsymbol{s}}=[\boldsymbol{s},\text{index}]$ is more expressive. We will focus on the actionable items you provide to increase our rating. Since the modification is slight in our code, we believe we can obtain results before the deadline. We will provide the results once we obtain them. Thanks a lot for your consideration.

---

> > > > > > > > > > ### Author Response · Authors · 2022-08-08
> > > > > > > > > > **Paper modifications, pseudo code, and new experimental results, part 1**
> > > > > > > > > >
> > > > > > > > > > We highly appreciate the reviewer's advice to smooth the transition of the paper, provide a pseudo code, and modify the model. We have worked heavily to achieve these three goals. Notably, the reviewer's insightful suggestion for the model finally leads to better performances. We are really thankful for these discussions.  We summarize our efforts as follows.
> > > > > > > > > >
> > > > > > > > > > $\textbf{Paper modifications.}$ We have two parts for the paper modification. First, we modified what we promised in the last response. Specifically, we add the descriptions of minimizing LoCaL in lines 161-165 and lines 217-218. Second, we follow the reviewer's summary and improve the organization of the paper. For example, we have the following improvements. $(i)$ We emphasize the optimization variable of LoCaL in lines 111-112. $(ii)$ We state that the multiplication layer of LoCaL does not have weights in line 120. $(iii)$ We state the search process is to identify $\boldsymbol{Z}_k$ in LoCaL in lines 126-128. $(iv)$ We define $\boldsymbol{s}_0$ in line 138 and the state-action trajectory in lines 144-146. $(v)$ We define the CMP in lines 147-151.
> > > > > > > > > >
> > > > > > > > > > $\textbf{Pseudo code.}$ We provide a high-level pseudo code to summarize our framework. The code is in Appendix A.1 and please refer to the revision for more details. In general, the code describes LoCaL structures and functions, the repeated search and estimation process, and the evaluation.
> > > > > > > > > >
> > > > > > > > > > $\textbf{New experiments.}$ First, we change the definitions of states by appending a state index entry to the original state vector. We show why and how to do the appending and define the state in lines 131-140. Second, we propose the new state transition process in Equation (1), which is still linear to support the proof of our Theorem 1. Third, we provide comparisons between the results before the state modification (OLD) and those after the state modification (NEW). We do not present the results of other methods due to space limits.
> > > > > > > > > >
> > > > > > > > > > $\textbf{Comparison 1. Learned equations.}$ For dataset $\text{Syn}_1$, the learned equations do not change.
> > > > > > > > > >
> > > > > > > > > > For dataset $\text{Syn}_2$, the learned equations (NEW) are: $y_1=1.48\sqrt{x_1}x_2+1.00x_1x\_2\^2$. $y_2=\sin{1.8x_1}(\log{2.999x_2}+0.999\sqrt{x_3})$. $y_3=1.933\sqrt{x_3}\log{1.598x_1}+1.002x\_1\^2$.
> > > > > > > > > >
> > > > > > > > > > For dataset $\text{Syn}_2$, the learned equations (OLD) are: $y_1=1.47\sqrt{x_1}x_2+1.001x_1x\_2\^2$. $y_2=\sin{1.8x_1}(\log{2.998x_2}+0.998\sqrt{x_3})$. $y_3=1.931\sqrt{x_3}\log{1.594x_1}+1.007x\_1\^2$.
> > > > > > > > > >
> > > > > > > > > > If we compare the two learned equations to the ground truth equations in lines 258-260, we find that the NEW results are more accurate in terms of parameters.
> > > > > > > > > >
> > > > > > > > > > $\textbf{Comparison 2. $E_c(\\%)$ for different datasets.}$ We define $E_c(\\%)$ as the average percentage errors in lines 297-302. Then, we have the following results.
> > > > > > > > > >
> > > > > > > > > > $E_c(\\%)$ $\\:$ $\\:$ $\\:$ $\\:$ $\text{Syn}_1$ $\\:$ $\\:$ $\\:$ $\\:$$\text{Syn}_2$ $\\:$ $\\:$ $\\:$ $\\:$$\text{Pow}$ $\\:$ $\\:$ $\\:$ $\\:$$\text{Mas}$
> > > > > > > > > >
> > > > > > > > > > NEW $\\:$ $\\:$ $\\:$ $\\:$ $\\:$ $\\:$ $0.03$ $\\:$ $\\:$ $\\:$ $\\:$ $\\:$ $0.19$ $\\:$ $\\:$ $\\:$ $\\:$ $1.76$ $\\:$ $\\:$ $\\:$ $\\:$ $1.28$
> > > > > > > > > >
> > > > > > > > > > OLD $\\:$ $\\:$ $\\:$ $\\:$ $\\:$ $\\:$  $0.03$ $\\:$ $\\:$ $\\:$ $\\:$ $\\:$ $0.38$ $\\:$ $\\:$ $\\:$ $\\:$ $2.53$ $\\:$ $\\:$ $\\:$ $\\:$ $1.48$
> > > > > > > > > >
> > > > > > > > > > $\textbf{Comparison 3. $\text{NRMSE}$ for different datasets.}$ We define $\text{NRMSE}$ as the normalized MSE in lines 164-165. Then, we have the following results.
> > > > > > > > > >
> > > > > > > > > > $\text{NRMSE}$ $\\:$ $\\:$ $\\:$ $\\:$ $\text{Syn}_1$ $\\:$ $\\:$  $\\:$  $\\:$ $\\:$ $\\:$ $\\:$ $\\:$ $\\:$ $\\:$ $\\:$ $\text{Syn}_2$ $\\:$ $\\:$ $\\:$ $\\:$ $\\:$ $\\:$  $\\:$ $\\:$  $\text{Pow}$ $\\:$ $\\:$  $\\:$ $\\:$  $\\:$ $\\:$ $\\:$ $\\:$$\text{Mas}$
> > > > > > > > > >
> > > > > > > > > > NEW $\\:$ $\\:$ $\\:$ $\\:$ $\\:$  $\\:$  $\\:$ $4.8\times 10^{-6}$ $\\:$ $\\:$ $\\:$ $\\:$ $\\:$ $5.2\times 10^{-6}$ $\\:$ $\\:$ $\\:$ $\\:$ $0.019$ $\\:$ $\\:$ $\\:$ $\\:$ $\\:$ $\\:$  $\\:$ $0.01$
> > > > > > > > > >
> > > > > > > > > > OLD $\\:$ $\\:$ $\\:$ $\\:$ $\\:$  $\\:$  $\\:$ $4.8\times 10^{-6}$ $\\:$ $\\:$ $\\:$ $\\:$ $\\:$ $7.5\times 10^{-6}$ $\\:$ $\\:$ $\\:$ $\\:$ $\\:$$0.022$ $\\:$ $\\:$ $\\:$ $\\:$ $\\:$ $\\:$  $\\:$ $0.015$

---

> > > > > > > > > > > ### Author Response · Authors · 2022-08-08
> > > > > > > > > > > **Paper modifications, pseudo code, and new experimental results, part 2**
> > > > > > > > > > >
> > > > > > > > > > > $\textbf{Comparison 4. $E_c(\\%)$ for Ablation Study.}$ The setting of the ablation study is in lines 253-268. Then, we have the following results.
> > > > > > > > > > >
> > > > > > > > > > > For dataset $\text{Syn}_1$, we have the following result.
> > > > > > > > > > >
> > > > > > > > > > >  $\\:$ $\\:$ $\\:$ $\\:$  $\\:$ No ablation   $\\:$ $\\:$  $\\:$ $\\:$  No exploration   $\\:$ $\\:$  $\\:$ $\\:$ No convex search   $\\:$ $\\:$  $\\:$ $\\:$ No LoCaL   $\\:$ $\\:$  $\\:$ $\\:$ No static const. $\\:$ $\\:$  $\\:$ $\\:$ No dynamic const.
> > > > > > > > > > >
> > > > > > > > > > > NEW $\\:$ $\\:$  $\\:$ $0.03$ $\\:$ $\\:$  $\\:$ $\\:$ $\\:$  $\\:$  $\\:$ $\\:$  $\\:$  $\\:$ $\\:$  $\\:$ $31.60$  $\\:$ $\\:$  $\\:$ $\\:$ $\\:$  $\\:$  $\\:$ $\\:$  $\\:$  $\\:$ $\\:$  $\\:$ $23.21$ $\\:$ $\\:$  $\\:$ $\\:$ $\\:$  $\\:$  $\\:$ $\\:$  $\\:$  $\\:$ $\\:$  $\\:$  $\\:$  $\\:$ $0.22$ $\\:$ $\\:$  $\\:$ $\\:$ $\\:$  $\\:$  $\\:$ $\\:$  $\\:$  $\\:$ $\\:$  $\\:$  $0.08$ $\\:$ $\\:$  $\\:$ $\\:$ $\\:$  $\\:$  $\\:$ $\\:$  $\\:$  $\\:$ $\\:$  $\\:$  $0.33$
> > > > > > > > > > >
> > > > > > > > > > > OLD $\\:$ $\\:$ $\\:$  $\\:$ $0.03$ $\\:$ $\\:$  $\\:$ $\\:$ $\\:$  $\\:$  $\\:$ $\\:$  $\\:$  $\\:$ $\\:$  $\\:$ $32.55$  $\\:$ $\\:$  $\\:$ $\\:$ $\\:$  $\\:$  $\\:$ $\\:$  $\\:$  $\\:$ $\\:$  $\\:$ $23.63$ $\\:$ $\\:$  $\\:$ $\\:$ $\\:$  $\\:$  $\\:$ $\\:$  $\\:$  $\\:$ $\\:$  $\\:$  $\\:$  $\\:$ $0.24$ $\\:$ $\\:$  $\\:$ $\\:$ $\\:$  $\\:$  $\\:$ $\\:$  $\\:$  $\\:$ $\\:$  $\\:$  $0.15$ $\\:$ $\\:$  $\\:$ $\\:$ $\\:$  $\\:$  $\\:$ $\\:$  $\\:$  $\\:$ $\\:$  $\\:$  $0.54$
> > > > > > > > > > >
> > > > > > > > > > > For dataset $\text{Syn}_2$, we have the following result.
> > > > > > > > > > >
> > > > > > > > > > >  $\\:$ $\\:$ $\\:$ $\\:$  $\\:$ No ablation   $\\:$ $\\:$  $\\:$ $\\:$  No exploration   $\\:$ $\\:$  $\\:$ $\\:$ No convex search   $\\:$ $\\:$  $\\:$ $\\:$ No LoCaL   $\\:$ $\\:$  $\\:$ $\\:$ No static const. $\\:$ $\\:$  $\\:$ $\\:$ No dynamic const.
> > > > > > > > > > >
> > > > > > > > > > > NEW $\\:$ $\\:$  $\\:$ $0.19$ $\\:$ $\\:$  $\\:$ $\\:$ $\\:$  $\\:$  $\\:$ $\\:$  $\\:$  $\\:$ $\\:$  $\\:$ $43.17$  $\\:$ $\\:$  $\\:$ $\\:$ $\\:$  $\\:$  $\\:$ $\\:$  $\\:$  $\\:$ $\\:$  $\\:$ $46.75$ $\\:$ $\\:$  $\\:$ $\\:$ $\\:$  $\\:$  $\\:$ $\\:$  $\\:$  $\\:$ $\\:$  $\\:$  $\\:$  $\\:$ $0.89$ $\\:$ $\\:$  $\\:$ $\\:$ $\\:$  $\\:$  $\\:$ $\\:$  $\\:$  $\\:$ $\\:$  $\\:$  $0.35$ $\\:$ $\\:$  $\\:$ $\\:$ $\\:$  $\\:$  $\\:$ $\\:$  $\\:$  $\\:$ $\\:$  $\\:$  $0.28$
> > > > > > > > > > >
> > > > > > > > > > > OLD $\\:$ $\\:$ $\\:$  $\\:$ $0.38$ $\\:$ $\\:$  $\\:$ $\\:$ $\\:$  $\\:$  $\\:$ $\\:$  $\\:$  $\\:$ $\\:$  $\\:$ $45.58$  $\\:$ $\\:$  $\\:$ $\\:$ $\\:$  $\\:$  $\\:$ $\\:$  $\\:$  $\\:$ $\\:$  $\\:$ $48.87$ $\\:$ $\\:$  $\\:$ $\\:$ $\\:$  $\\:$  $\\:$ $\\:$  $\\:$  $\\:$ $\\:$  $\\:$  $\\:$  $\\:$ $1.05$ $\\:$ $\\:$  $\\:$ $\\:$ $\\:$  $\\:$  $\\:$ $\\:$  $\\:$  $\\:$ $\\:$  $\\:$  $0.87$ $\\:$ $\\:$  $\\:$ $\\:$ $\\:$  $\\:$  $\\:$ $\\:$  $\\:$  $\\:$ $\\:$  $\\:$  $0.52$
> > > > > > > > > > >
> > > > > > > > > > > $\textbf{Comparison 5. $E_c(\\%)$ for Sensitivity Analysis with respect to noises.}$ The setting is in lines 371-377. Then, we have the following results.
> > > > > > > > > > >
> > > > > > > > > > > For dataset $\text{Syn}_1$, we have the following result.
> > > > > > > > > > >
> > > > > > > > > > > SNR $\\:$ $\\:$ $\\:$ $\\:$  $\\:$  $\\:$ $\\:$ $80$ $\\:$ $\\:$ $\\:$ $\\:$ $\\:$ $\\:$  $\\:$  $\\:$ $90$  $\\:$ $\\:$ $\\:$ $\\:$ $\\:$ $\\:$  $\\:$  $\\:$ $100$  $\\:$ $\\:$ $\\:$ $\\:$ $\\:$ $\\:$  $\\:$  $\\:$ $110$ $\\:$ $\\:$ $\\:$ $\\:$ $\\:$ $\\:$  $\\:$  $\\:$ $120$
> > > > > > > > > > >
> > > > > > > > > > > NEW $\\:$ $\\:$ $\\:$ $\\:$  $\\:$  $\\:$ $4.21$ $\\:$ $\\:$ $\\:$ $\\:$  $\\:$  $\\:$ $1.08$ $\\:$ $\\:$ $\\:$  $\\:$  $\\:$ $\\:$ $\\:$ $0.20$ $\\:$ $\\:$ $\\:$ $\\:$  $\\:$  $\\:$ $\\:$  $0.14$ $\\:$ $\\:$ $\\:$ $\\:$  $\\:$  $\\:$ $\\:$  $0.03$
> > > > > > > > > > >
> > > > > > > > > > > OLD $\\:$ $\\:$ $\\:$ $\\:$  $\\:$  $\\:$ $\\:$  $5.23$ $\\:$ $\\:$ $\\:$  $\\:$  $\\:$ $\\:$ $2.35$ $\\:$ $\\:$ $\\:$  $\\:$  $\\:$ $\\:$ $\\:$ $0.29$ $\\:$ $\\:$ $\\:$ $\\:$  $\\:$  $\\:$ $\\:$  $0.16$ $\\:$ $\\:$ $\\:$ $\\:$  $\\:$  $\\:$ $\\:$  $0.04$
> > > > > > > > > > >
> > > > > > > > > > > For dataset $\text{Syn}_2$, we have the following result.
> > > > > > > > > > >
> > > > > > > > > > > SNR $\\:$ $\\:$ $\\:$ $\\:$  $\\:$  $\\:$ $\\:$ $80$ $\\:$ $\\:$ $\\:$ $\\:$ $\\:$ $\\:$  $\\:$  $\\:$ $90$  $\\:$ $\\:$ $\\:$ $\\:$ $\\:$ $\\:$  $\\:$  $\\:$ $100$  $\\:$ $\\:$ $\\:$ $\\:$ $\\:$ $\\:$  $\\:$  $\\:$ $110$ $\\:$ $\\:$ $\\:$ $\\:$ $\\:$ $\\:$  $\\:$  $\\:$ $120$
> > > > > > > > > > >
> > > > > > > > > > > NEW $\\:$ $\\:$ $\\:$ $\\:$  $\\:$  $\\:$ $4.03$ $\\:$ $\\:$ $\\:$ $\\:$  $\\:$  $\\:$ $2.75$ $\\:$ $\\:$ $\\:$  $\\:$  $\\:$ $\\:$ $\\:$ $0.56$ $\\:$ $\\:$ $\\:$ $\\:$  $\\:$  $\\:$ $\\:$  $0.45$ $\\:$ $\\:$ $\\:$ $\\:$  $\\:$  $\\:$ $\\:$  $0.33$
> > > > > > > > > > >
> > > > > > > > > > > OLD $\\:$ $\\:$ $\\:$ $\\:$  $\\:$  $\\:$ $\\:$  $4.79$ $\\:$ $\\:$ $\\:$  $\\:$  $\\:$ $\\:$ $3.59$ $\\:$ $\\:$ $\\:$  $\\:$  $\\:$ $\\:$ $\\:$ $0.91$ $\\:$ $\\:$ $\\:$ $\\:$  $\\:$  $\\:$ $\\:$  $0.88$ $\\:$ $\\:$ $\\:$ $\\:$  $\\:$  $\\:$ $\\:$  $0.50$

---

> > > > > > > > > > > > ### Author Response · Authors · 2022-08-08
> > > > > > > > > > > > **Paper modifications, pseudo code, and new experimental results, part 3**
> > > > > > > > > > > >
> > > > > > > > > > > > $\textbf{Comparison 6. $E_c(\\%)$ for Sensitivity Analysis with respect to sample numbers.}$ The setting is in lines 371-377. Then, we have the following results.
> > > > > > > > > > > >
> > > > > > > > > > > > For dataset $\text{Syn}_1$, we have the following result.
> > > > > > > > > > > >
> > > > > > > > > > > > Sample Number $\\:$ $\\:$ $\\:$  $\\:$ $\\:$ $\\:$ $500$ $\\:$ $\\:$ $\\:$  $\\:$ $\\:$ $\\:$ $1000$ $\\:$ $\\:$ $\\:$  $\\:$ $\\:$ $\\:$ $1500$ $\\:$ $\\:$ $\\:$  $\\:$ $\\:$ $\\:$ $2000$ $\\:$ $\\:$ $\\:$  $\\:$ $\\:$ $\\:$ $2500$
> > > > > > > > > > > >
> > > > > > > > > > > > NEW $\\:$ $\\:$ $\\:$  $\\:$ $\\:$ $\\:$ $\\:$ $\\:$ $\\:$  $\\:$ $\\:$ $\\:$ $\\:$  $\\:$ $\\:$ $\\:$ $0.54$ $\\:$  $\\:$ $\\:$  $\\:$ $\\:$ $\\:$ $0.34$ $\\:$  $\\:$ $\\:$  $\\:$ $\\:$ $\\:$ $0.26$ $\\:$  $\\:$ $\\:$  $\\:$ $\\:$ $\\:$  $\\:$ $0.20$ $\\:$  $\\:$  $\\:$ $\\:$  $\\:$ $\\:$ $\\:$ $0.18$
> > > > > > > > > > > >
> > > > > > > > > > > > OLD $\\:$ $\\:$ $\\:$  $\\:$ $\\:$ $\\:$ $\\:$ $\\:$ $\\:$  $\\:$ $\\:$ $\\:$ $\\:$  $\\:$ $\\:$ $\\:$ $0.86$ $\\:$  $\\:$ $\\:$  $\\:$ $\\:$ $\\:$ $0.63$ $\\:$  $\\:$ $\\:$  $\\:$ $\\:$ $\\:$ $0.45$ $\\:$  $\\:$ $\\:$  $\\:$ $\\:$ $\\:$  $\\:$  $\\:$ $0.29$ $\\:$  $\\:$  $\\:$ $\\:$  $\\:$ $\\:$ $\\:$  $0.25$
> > > > > > > > > > > >
> > > > > > > > > > > > For dataset $\text{Syn}_2$, we have the following result.
> > > > > > > > > > > >
> > > > > > > > > > > > Sample Number $\\:$ $\\:$ $\\:$  $\\:$ $\\:$ $\\:$ $500$ $\\:$ $\\:$ $\\:$  $\\:$ $\\:$ $\\:$ $1000$ $\\:$ $\\:$ $\\:$  $\\:$ $\\:$ $\\:$ $1500$ $\\:$ $\\:$ $\\:$  $\\:$ $\\:$ $\\:$ $2000$ $\\:$ $\\:$ $\\:$  $\\:$ $\\:$ $\\:$ $2500$
> > > > > > > > > > > >
> > > > > > > > > > > > NEW $\\:$ $\\:$ $\\:$  $\\:$ $\\:$ $\\:$ $\\:$ $\\:$ $\\:$  $\\:$ $\\:$ $\\:$ $\\:$  $\\:$ $\\:$ $\\:$ $1.15$ $\\:$  $\\:$ $\\:$  $\\:$ $\\:$ $\\:$ $0.91$ $\\:$  $\\:$ $\\:$  $\\:$ $\\:$ $\\:$ $0.72$ $\\:$  $\\:$ $\\:$  $\\:$ $\\:$ $\\:$  $\\:$ $0.56$ $\\:$  $\\:$  $\\:$ $\\:$  $\\:$ $\\:$ $\\:$ $0.45$
> > > > > > > > > > > >
> > > > > > > > > > > > OLD $\\:$ $\\:$ $\\:$  $\\:$ $\\:$ $\\:$ $\\:$ $\\:$ $\\:$  $\\:$ $\\:$ $\\:$ $\\:$  $\\:$ $\\:$ $\\:$ $1.32$ $\\:$  $\\:$ $\\:$  $\\:$ $\\:$ $\\:$ $1.14$ $\\:$  $\\:$ $\\:$  $\\:$ $\\:$ $\\:$ $0.92$ $\\:$  $\\:$ $\\:$  $\\:$ $\\:$ $\\:$  $\\:$  $\\:$ $0.91$ $\\:$  $\\:$  $\\:$ $\\:$  $\\:$ $\\:$ $\\:$  $0.74$
> > > > > > > > > > > >
> > > > > > > > > > > > From the above comparisons, we have the following observations. First, the modified states in the NEW model improve the search process and bring lower errors. The key is that the state modification can help to remove the scenario of duplicated states in one trajectory. This encourages the sequential decision-making to consider the state index information (i.e., the depth of the layer in LoCaL) to achieve fast convergence. It also avoids searching for different optimal actions of one state. Second, there are scenarios when the NEW model has very slight improvements compared to the OLD model. In these scenarios, the search space is small and the exploration strategy (i.e., $\epsilon$-greedy strategy) can help to guarantee the convergence even for the OLD model.
> > > > > > > > > > > >
> > > > > > > > > > > > Finally, we utilize values of the NEW model to plot figures and update Fig. 3-5 and Table 1 in the revised paper. Please refer to the revision for the new figures. We hope the above responses are satisfactory to the reviewer. We are really grateful for the reviewer's efforts to make the model sound and effective.

---

### Author Response · Authors · 2022-08-02
**Rebuttal Summary**

Thanks a lot for the reviewers' questions. We have addressed all the comments. Our key effort is to remove the theoretical flaws mentioned by Reviewer 1. We also make clarifications for some factual misunderstandings and correct typos. Please see the revised paper with modifications marked in blue. As for specific questions, please see our comments. Thanks for considering our paper.

---

### Comment · Reviewer_AUni · 2022-08-08
**Thank you for re-runing experiments**

I'd like to thank the authors for the important work they put during the rebuttal. I believe that quality of the paper has been improved a lot since the initial submission. I am particularly happy to see the experimental results have improved.

Note that with state index it is now a MDP again (with non zero reward received at states of index K+1) so Split-POMDP machinery may not be necessary anymore. But I don't want to be picky !

I do not have any remaining remark related to soundness or clarity of the paper. As promised, I increase my rating to reflect my new opinion on the current state of the manuscript. Congrats !

---

> ### Author Response · Authors · 2022-08-09
> **Thanks a lot for your appreciation**
>
> We are really grateful for your constructive comments that largely increase the effectiveness and soundness of our work. We really enjoy the discussion with you these days. Thanks a lot for your consideration and appreciation.

---

### Meta-Review · Area_Chair_Ap7Y · 2022-08-31

**Recommendation:** Accept
**Confidence:** Less certain

**Metareview:**

Authors propose to learn coefficients and symbols connections in the problem of Symbolic Regression (SR) by combining Q learning and neural networks. They rely on the convexity properties of of ICNN to ensure that the Q function and the rewards R() are convex wrt the state S and the action A of the MDP defined. They use the convexity properties and the determinism of the MDP to derive several guarantees on their method - namely, that the exact symbols can be recovered for noiseless datasets because the problem is locally convex in a neighborhood of the true minimum.

There were a number of questions and concerns regarding the theoretical results raised the reviewers which were addressed during the discussion period. I'd like to thank the authors for the important work they put during the rebuttal. The quality of the paper has been improved a lot since the initial submission and the experimental results have improved.

**Award:**

No

---

### Decision · Program_Chairs · 2022-09-14

Accept